# Laser scribed proton exchange membranes for enhanced fuel cell performance and stability

Jianuo Chen [1,2], Xuekun Lu [3], Lingtao Wang[4], Wenjia Du [1,5], Hengyi Guo[4], Max Rimmer [4], Heng Zhai [2], Yuhan Liu[1], Paul R. Shearing[5], Sarah J. Haigh [4], Stuart M. Holmes[2] & Thomas S. Miller [1] ✉

High-temperature proton exchange membrane fuel cells (HT-PEMFCs) offer solutions to challenges intrinsic to low-temperature PEMFCs, such as complex water management, fuel inflexibility, and thermal integration. However, they are hindered by phosphoric acid (PA) leaching and catalyst migration, which destabilize the critical three-phase interface within the membrane electrode assembly (MEA). This study presents an innovative approach to enhance HT-PEMFC performance through membrane modification using picosecond laser scribing, which optimises the three-phase interface by forming a graphene-like structure that mitigates PA leaching. Our results demonstrate that laser-induced modification of PA-doped membranes, particularly on the cathode side, significantly enhances the performance and durability of HT-PEMFCs, achieving a peak power density of 817.2 mW cm$^{-2}$ after accelerated stress testing, representing a notable 58.2% increase compared to untreated membranes. Furthermore, a comprehensive three-dimensional multi-physics model, based on X-ray micro-computed tomography data, was employed to visualise and quantify the impact of this laser treatment on the dynamic electrochemical processes within the MEA. Hence, this work provides both a scalable methodology to stabilise an important future membrane technology, and a clear mechanistic understanding of how this targeted laser modification acts to optimise the three-phase interface of HT-PEMFCs, which can have impact across a wide array of applications.

The performance of proton exchange membrane fuel cells (PEMFCs) is defined by a series of intricate interactions involving three phases; the migration and distribution of each phase have significant implications for overall device performance and durability[1,2]. While proton exchange membranes (PEMs) and catalysts are widely recognised to define the performance of PEMFCs and hence have been the focal point of extensive research[3,4], the importance of achieving cross-component synergy through design optimization or intervention strategies should not be understated. In the case of the most typical low-temperature PEMFCs, which are based on perfluorosulfonic acid membranes, effective water management emerges as a primary factor for enhancing fuel cell performance[5-7]. Unfortunately managing water adds significantly to the system cost and complexity. High-temperature proton exchange membrane fuel cells (HT-PEMFCs)

[1]Department of Chemical Engineering, Electrochemical Innovation Lab, University College London, London, UK. [2]Department of Chemical Engineering, University of Manchester, Manchester, UK. [3]School of Engineering and Materials Science, Queen Mary University of London, London, UK. [4]Department of Materials, University of Manchester, Manchester, UK. [5]Department of Engineering Science, University of Oxford, Oxford, UK. ✉e-mail: t.miller@ucl.ac.uk

operate at temperatures exceeding 100 °C, effectively removing the issue of water management while reducing the stringent requirement for high-purity hydrogen and providing a route to utilise waste heat. HT-PEMFCs commonly utilise phosphoric acid (PA) based membranes, which offer advantages including a high boiling point and good proton conductivity[8,9], however the migration of PA from the membrane into the catalyst layer, a phenomenon known as leaching, results in an uncertain distribution of the electrolyte and introduces complexity to the vitally important three-phase boundary. This can make the mass transfer, momentum transfer and the distribution of electrochemical reactions in HT-PEMFCs inconsistent, which can drive degradation[10–13].

The three-phase interface in HT-PEMFC serves as a critical locus for all of the key electrochemical processes in a fuel cell, acting as the site for oxygen reduction (at the cathode) and hydrogen oxidation (at the anode), meaning mass transport, as well as electrical and ionic conductivity, towards/from this boundary defines cell performance[14]. Hence, investigating these interfaces is highly complex, as the behavior of numerous phenomena, including electrochemistry[15], mass transport and fluid dynamics (free and porous media flow)[16,17], heat transfer (solid-state, fluidic, and electromagnetic heat)[18], as well as structural mechanics (membranes, porous elasticity)[19], must be simultaneously considered. Current research on the three-phase interface predominantly emphasises modifications to catalysts, binders, or flow fields on the electrodes[20–22], however the justification for these modifications is commonly based on little more than 'trial-and-error' approaches and the assessment of their success is usually decided simply by ultimate fuel cell current or power density, or durability.

One area of particular research focus for HT-PEMFCs is the development of membrane systems. Various polymers have been developed to enhance membrane proton conductivity of HT-PEMFCs, such as polyethersulfone–polyvinylpyrrolidone (PES–PVP)[23], while others, like polymers of intrinsic microporosity (PIMs)[24], have been employed to improve the longevity of HT-PEMFCs under complex conditions. However, advancements in the most developed polymer for HT-PEMFCs, polybenzimidazole (PBI), remains a focal point in this field. Modifications of PBI, including functionalization[25–27], cross-linking, grafting[28], and doping[29], have been extensively explored. HT-PEMFCs utilizing different membranes generally achieve peak power densities in the range of 400–700 mW cm$^{-2}$[23,30]. However, these membranes typically exhibit varying degrees of performance degradation during durability tests[26,28,29], with only a few studies demonstrating stable performance[31]. Unfortunately, as the membrane electrode assembly (MEA) is so complex, it is often impossible to determine if it is the intended change (such as a new polymer membrane or catalyst) to the MEA or other secondary influences (such as improved ionomer distribution caused by changes to the manufacturing process needed to integrate the new catalyst) that are truly driving the fuel cell to perform better. Only tools that enable us to 'look inside' operating MEAs will allow these influences to be deconvoluted.

The utilization of image-based multi-physics simulations has emerged as a powerful tool for the integrated analysis of multiple components in complex systems and the optimization of electrochemical devices. Nevertheless, due to the complexity of MEAs in fuel cells, contemporary simulations of PEMFCs have predominantly been confined to idealized two-dimensional or three-dimensional models based on homogeneous descriptions of MEA materials[32–35]. Furthermore, where three-dimensional modelling has been used, it has commonly emphasized single-physics fluid simulations within the gas channels or simplified multi-physics simulations, excluding consideration of electrochemical processes, which neglects the critical interaction between fluid dynamics and electrochemical processes[33,36,37]. Importantly, X-ray micro-computed tomography (μ-CT), coupled with machine learning segmentation, has nonetheless enabled the non-destructive visualisation of full MEAs and the ability to distinguish between the different components and has proven effective for experimental characterization of catalysts, PA migration, and porous media porosity in HT-PEMFCs[10]. Hence, μ-CT can offer a route towards the authentic representation of MEA structures for multi-physics coupling simulations of fuel cells and enable the development of accurate descriptions of the behavior of the triple-phase boundary, membrane, and beyond while under electrochemical control. This has yet to be explored in the academic literature.

In this work, we will combine advanced characterization, via X-ray μ-CT and image-based modelling to develop a visualization model of dynamic electrochemical processes based on real MEA structure to enable a real understanding of the influence of MEA materials and method of manufacturing on electrochemical performance. Then through the application of this model and a combination of next-generation manufacturing and materials engineering, we will develop a new, durable, high-performance, and most importantly scalable HT-PEMFC MEA system based on a membrane modified with graphene synthesised in-situ through the use of laser scribing.

Graphene and graphene oxide (GO) have both been shown to demonstrate high proton conductivity[38–41]. Additionally, single-layer graphene (SLG), with its unique two-dimensional hexagonal structure, has demonstrated an ability to not only facilitate the passage of protons but also effectively hinder the permeation of PA and hydrogen gas[42–45]. While graphene-related materials have found widespread applications as dopants in membranes and catalyst substrates for PEMFCs[46–49], their utilization between the PEM and the catalyst interface is much rarer, often confined to theoretical exploration[44,45,50]. In the only example of this methodology for HT-PEMFCs, the graphene used was prepared through chemical vapour deposition (CVD) and wet chemical transfer, which both have significant barriers for industrial application[51]. CVD offers the capability to produce high-quality SLG, however, its scalability and commercialization are constrained by elevated costs, intricate processes, and concerns related to safety and environmental impact[52–54]. Additionally, the wet chemical transfer method involves multiple immersions of the substrate in solvents, which makes the process difficult to scale, specifically for HT-PEMFC membranes leading to the potential loss of PA within the membrane[55]. Consequently, this method is more suitable for transferring graphene onto electrode surfaces rather than membranes, but the high roughness of the electrode surface results in graphene fracturing after hot pressing, causing uncontrollable coverage between the membrane and the catalyst[51].

In contrast to CVD, it has also been shown that graphene can be produced directly on commercially available polymer films via a one-step laser scribing method, although here the graphene structure was highly wrinkled and puckered[56]. This laser-induced graphene (LIG) offers advantages in terms of scalability, cost-efficiency, and the capability to pattern and manufacture large-area devices, thus holding potential for commercial production[57,58]. LIG offers a significantly increased surface area and conductivity, compared to traditional graphene preparation methods including CVD and exfoliation, which has been shown to be advantageous in energy storage applications[56,58]. For example, Luo et al. have demonstrated that using PA-doped polymer as a precursor for laser-induced modification can aid in generating porous graphene materials that enhance ion migration in supercapacitors[59]. Fast ion transport is vital for HT-PEMFC membranes, suggesting this method could offer significant advantages in this application.

Hence, as a demonstration of the power of image-based modelling to deconvolute the influence of novel materials and manufacturing on the performance of an MEA, we will treat PA-doped PBI membranes with ultraviolet picosecond laser surface treatment to create a hybrid membrane system to both hinder PA leaching and reactant crossover, offering high stability HT-PEMFC MEAs.

## Results and discussion
### PA doping and laser-induced modification of membranes
PBI necessitates immersion in PA at specific temperatures for durations to achieve PA-doped PBI. The PBI membranes, both before and

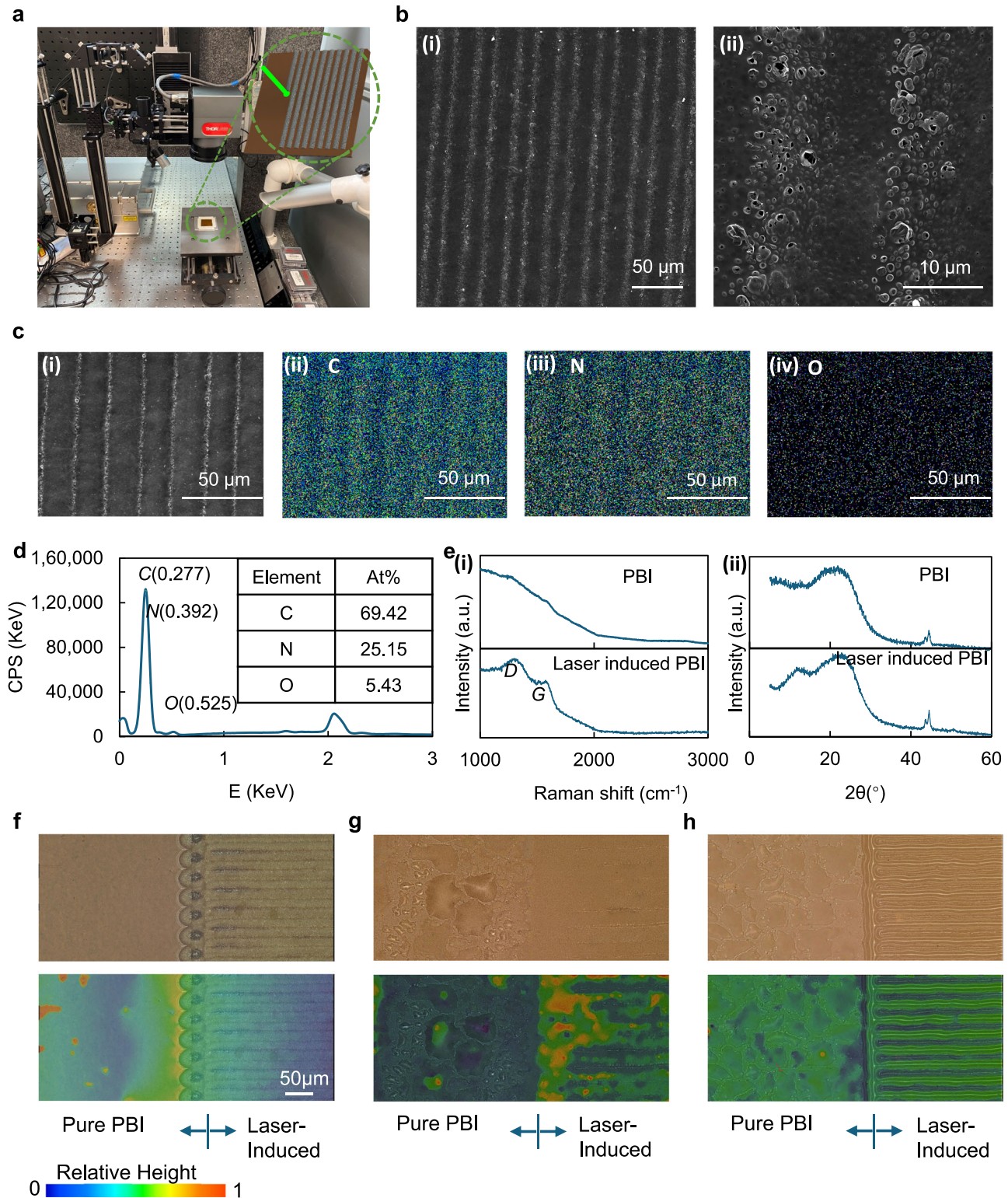

**Fig. 1 | Laser-induced membrane. a** Laser induction set up. **b** SEM images (i) large scale and (ii) small scale. **c** EDX images, (i) SEM image for EDX, (ii) C element, (iii) N element and (iv) O element, **d** EDX spectrum and elements content, **e** (i) Raman spectra of PBI membrane and Laser-induced PBI membrane, (ii) XRD spectra of PBI membrane and Laser-induced PBI membrane, **f** optical microscopy image and depth profile of laser-induced PBI membrane, **g** optical microscopy and depth profile of PA doped Laser-induced PBI membrane, and **h** optical microscopy and depth composition profile of laser-induced PA doped PBI membrane.

after PA doping, were subjected to line-by-line scanning under pico-second laser exposure, as illustrated in Fig. 1a. To investigate the effects of laser irradiation on PBI membranes, Scanning Electron Microscopy (SEM), energy-dispersive x-ray spectroscopy (EDX) and Raman spectroscopy were applied to explore the changes in morphology and chemical composition of the membranes. SEM images and an EDX map of a laser-induced PBI membrane without PA doping are shown in Fig. 1b, c. An optical microscopy depth profile is shown in Fig. S1. The regions traversed by the center of the laser spot exhibited distinct differences from those that interacted with the spot

edges. This leads to localized differentiation in the distribution of PA, thereby affecting the three-phase interface. In the SEM images, this is manifested as brightened areas, and upon further magnification, microscale thin film foam-like structures become observable, consistent with previous demonstrations of LIG. Similar structures can be seen between the areas that experienced the peak laser intensity, but with a lower distribution. EDX mapping (Fig. 1c) revealed that the distribution of C and N content, from the PBI framework at the laser footprint, becomes sparser, accompanied by the presence of oxygen, an element not inherent to the PBI framework. Additionally, as evident from the EDX spectrum in Fig. 1d, which represents a typical spectrum obtained at the laser focal point, the O content remains substantial, with an atomic fraction of 5.43 At%. This clearly indicates that laser treatment induces both foaming and oxidation in PBI, Further validation of this can be gleaned from the Raman spectra in Fig. 1e (i). Due to the pronounced background fluorescence effect of PBI in Raman spectroscopy the Raman spectrum of the untreated membrane showed no significant features. This observation is attributed to the use of relatively high laser energy, specifically employed to enhance the visibility and characterization of graphene. Conversely, the Raman spectra at the laser footprints of laser-induced PBI revealed distinct $D$ (~1350 cm$^{-1}$) and $G$ (~1580 cm$^{-1}$) peaks, indicative of GO features, at their respective positions[60–62]. X-ray diffraction is a powerful technique used to characterize and differentiate graphite, GO, and graphene-based on their unique structural features. GO exhibits a distinct peak at around $2\theta \approx 10$–$12°$. This shift to a lower angle compared to graphite is due to the increased interlayer spacing caused by the presence of oxygen-containing functional groups. The increased spacing disrupts the regular stacking order seen in graphite. As shown in Fig. 1e (ii), compared to PBI, the laser-induced PBI displays a characteristic peak of GO at $2\theta = 11.68°$, providing strong evidence of the formation and presence of GO. This result, combined with the aforementioned Raman analyses, further substantiates the successful generation of GO on the membrane surface.

After PA doping of the PBI membrane, characterization was undertaken using optical microscopy, as the liquid contained within the membrane ruled out many other techniques. Optical microscopy images and depth composition profiling of laser-induced PBI, PA-doped laser-induced PBI, and laser-induced PA-doped PBI are shown in Fig. 1f–h, respectively. PA-doped laser-induced PBI denotes the process where the PBI membrane underwent laser induction prior to PA doping, whereas laser-induced PA-doped PBI signifies the material where PBI was first doped with PA and subsequently subjected to laser induction. From Fig. 1f, it is evident that there were distinct characteristic differences between the pure PBI membrane and the PBI membrane subjected to laser treatment. The surface of the pure PBI membrane appears smooth, whereas the laser-treated PBI membrane exhibits noticeable protrusions due to graphene foam formation at the laser tracks, likely caused by the detachment of graphene oxide layers. From the optical microscope image in Fig. 1g, it can be observed that the surface of PA-doped PBI is covered with numerous droplets. This results from the precipitation of PA or the absorption of water vapour by PA from the air. In contrast, PA-doped laser-induced PBI shows no significant droplets. It therefore implies that the surface structure of laser-induced graphene oxide effectively hinders the leaching of PA and its contact with moisture from air. Moreover, due to the incorporation of PA, the laser-induced footprint features become less distinct compared to a pure PBI membrane. However, in the case of the laser-induced PA-doped membrane, the laser-induced footprints on the PA-doped membrane are more pronounced than those on the dry pure PBI membrane as shown in Fig. 1h. This can be attributed to the involvement of PA, which enhances the porous structure of laser-induced graphene, making the laser footprints more textured in the presence of PA doping. Although the surface gridding increases the roughness or unevenness of the membrane surface, it is expected to

**Table 1 | The nomenclature and membrane conditions for MEAs based on different membranes**

| Name | Laser processing | The position of laser processing |
|------|------------------|----------------------------------|
| PBI | Pure PBI | Pure PBI membrane-based MEA |
| La | Before PA doping | On the membrane anode side |
| Lc | Before PA doping | On the membrane cathode side |
| Lac | Before PA doping | On the membrane both sides |
| LPAa | After PA doping | On the membrane anode side |
| LPAc | After PA doping | On the membrane cathode side |
| LPAac | After PA doping | On the membrane both sides |

enhance the three-phase interface between the membrane, catalyst, and fuel for a given electrode area.

## Electrochemical performance, durability and characterization of MEAs

In order to investigate the influence of laser-induced effects on the electrochemical performance and durability of HT-PEMFC, polarization curves before and after accelerated stress testing (AST), as well as EIS before and after ASTs were conducted. The nomenclature and membrane preparation conditions for the different MEAs tested are presented in Table 1. The corresponding results from MEAs containing PBI, LPAa, LPAc, and LPAac membranes are illustrated in Fig. 2, panels a–e, while the results of La, Lc and Lac are shown in Fig. S2. The AST curves depicted in Figs. 2a and S2a reveal distinct differences in MEA performance, resulting in significant differences in voltage at various current densities before and after AST. PBI demonstrates a baseline OCV stability at ~1 V. However, both Lc and Lac exhibit poor initial OCV, approximately around 0.9 V. While La initially demonstrates an OCV of around 1 V, it also decays to ~0.95 V after AST. In contrast, LPAa, LPAc and LPAac exhibit a similarly stable OCV of ~1 V, mirroring the behavior observed in PBI. OCV reflects the crossover of hydrogen gas in fuel cells by indicating the voltage generated when no current flows through the system. An increase in hydrogen crossover leads to a decrease in OCV due to electrochemical reactions consuming hydrogen at the electrode surfaces, thereby reducing the voltage output. Therefore, it can be seen that performing PA doping prior to laser induction is more effective in reducing the impact of laser induction on gas crossover compared to doping after laser induction. This conclusion is further validated by the linear sweep voltammetry (LSV) curves of Lac and LPAac in Fig. S3. The calculated hydrogen crossover values for Lac and LPAac according to LSV are $0.71 \times 10^{-8}$ and $1.53 \times 10^{-8}$, respectively. Therefore, this study will primarily focus on the analysis of membranes subjected to PA doping followed by laser induction (LPAa, LPAc, and LPAac). During the AST process at various current densities, PBI exhibited varying degrees of voltage decline. In contrast, LPAc showed varying degrees of voltage increase, while the voltage of LPAa and LPAac remains relatively stable. The trends in variation during the initial six hours of AST (highlighted in the blue box in Fig. 2a) and the final six hours of AST (highlighted in the red box) are also indicative.

To illustrate the differences between various MEAs during the AST process, stacked AST plots from the first cycle to the fifth cycle, with each cycle spanning six hours, are shown in Fig. S4. As seen in Fig. S4, the performance of LPAa in the fifth cycle began to decline compared to the fourth cycle, whereas LPAc remained relatively stable. This further demonstrates the superiority of LPAc over LPAa.

The pre-and post-AST power density curve alterations align with the behavior observed in the polarisation curves before and after ASTs, depicted in Fig. 2b, c, respectively. LPAc consistently exhibited superior performance compared to all others both before and after AST. For example, its peak power density increased from 734.3 mW cm$^{-2}$ to 817.2 mW cm$^{-2}$, in sharp contrast to PBI, which

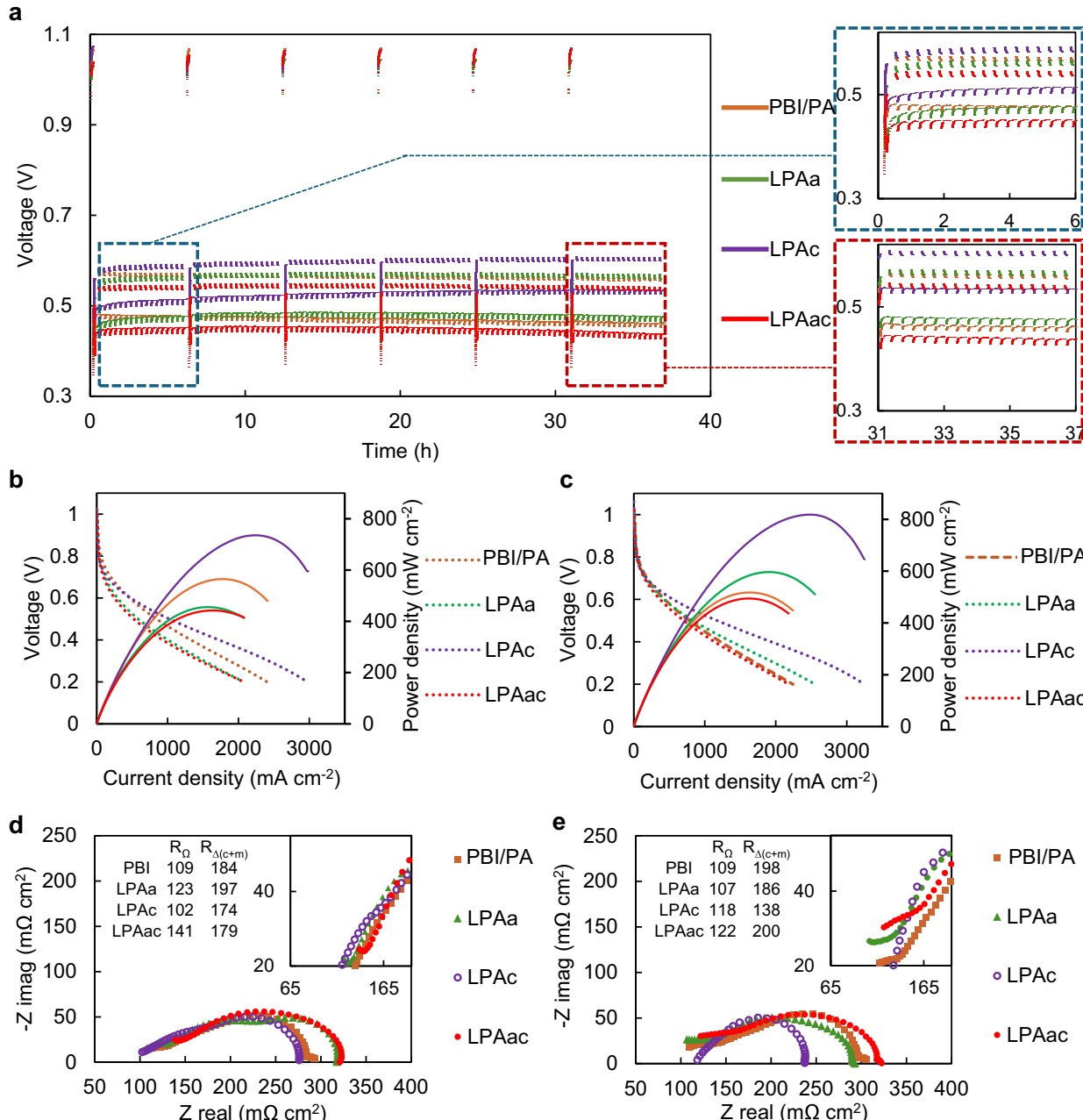

**Fig. 2 | Electrochemical characterization of MEAs based on different membranes 160°C, anode: H$_2$ ($\lambda$ = 1.2), cathode: O$_2$ ($\lambda$ = 2.0). a** AST (repeating chronopotentiometry between 0.6 A cm$^{-2}$ for 4 min and 1 A cm$^{-2}$ for 16 min, running OCV for 10 min every 6 hr), **b** polarization curves before AST, **c** polarization curves after AST, **d** EIS curves before AST (Nyquist plots, DC: 0.5 A cm$^{-2}$, frequency:10 kHz−0.1 Hz), **e** EIS curves after AST (Nyquist plots, DC: 0.5 A cm$^{-2}$, frequency:10 kHz−0.1 Hz).

experienced a notable decrease from 536.9 mW cm$^{-2}$ to 517.3 mW cm$^{-2}$. LPAa and LPAac both had lower peak power performance than PBI before the AST, but after 6 AST cycles, LPAa demonstrated a significant increase in peak power (from 442 mW cm$^{-2}$ to 596 mW cm$^{-2}$) while that of PBI dropped. While LPAac's performance never surpassed PBI during the testing period, its peak power density also improved during the AST, with a value close to that of PBI after AST. The extended AST plots of LPAc, as shown in Fig. S5, demonstrate that the performance of LPAc remains highly stable even up to 111 h. Different from LPAa, LPAc and LPAac, MEAs based on membranes treated with laser induction before PA doping (La, Lc, and Lac) consistently demonstrated worse performance than PBI's inferior performance overall. The incorporation of

PA doping prior to laser irradiation appears to be more conducive to the enhancement of performance and durability in HT-PEMFCs.

EIS can provide valuable information about the electrochemical processes occurring in fuel cells, including electrode kinetics, mass transport phenomena, and proton conductivity. It enables the characterization of the cell's performance, impedance, and overall electrochemical behavior, offering insights into factors affecting performance and durability. As illustrated in Fig. 2d, e and Fig. S2d, e, the Nyquist plots from different MEAs exhibit characteristic differences, particularly at high frequencies, La, Lac, LPAa, and LPAac all display an incomplete semicircle. The equivalent circuit used for fitting, and the parameters used, are shown in Table S1. The Ohmic

component ($R_\Omega$) and the sum of the charge transfer and mass transfer resistances on the cathode and anode $\left(R_{\Delta(c+m)}\right)$, provided a basis for comparison. From this analysis, several patterns become apparent. First, in the case of PBI, $R_\Omega$ was largely unchanged before and after AST, while $R_{\Delta(c+m)}$ experienced a noticeable increase, which is likely a primary contributing factor to the gradual decline in PBI performance. Secondly, MEAs involving laser treatment on the anode side of the membrane, including La, Lac, LPAa, and LPAac, exhibited a drop in $R_\Omega$ after AST. Conversely, MEAs involving laser treatment on cathode side of the membrane, Lc and LPAc, experienced an increase in $R_\Omega$. The average proton conductivity of different membranes is shown in Fig. S6. Regardless of whether laser treatment was applied before or after PA doping, the proton conductivity of the membranes did not improve compared to untreated PBI, generally a lower conductivity was observed. Membranes that underwent double-sided laser treatment exhibited lower conductivity compared to those treated on a single side. Importantly however, membranes subjected to laser treatment after PA doping demonstrated higher proton conductivity than those treated with laser before PA doping. Finally, for most laser-treated MEAs, except for Lc and LPAac, $R_{\Delta(c+m)}$ decreased after AST and was consistently lower than that of PBI. This indicates that laser-induced treatment results in a more rational distribution of PA within the electrode, consequently enhancing electrode kinetics. Notably, the MEA with the most optimal performance, LPAc, demonstrates the most significant reduction. It is evident that LPAc exhibits optimal performance and durability primarily due to the improved internal optimization of the electrode structure. Hence, these data suggest that laser treatment appears to somewhat compromise the initial performance of HT-PEMFC, but as complex processes unfold during AST, such as the redistribution of electrolytes, changes in membrane morphology, variations in conductivity, fluctuations in the three-phase interface, and variations in catalyst activity, one side laser treated membranes (particularly those with graphene on the cathode side) gradually enhances the electrochemical reaction performance within the electrode, consequently improving the performance and durability of the MEA, despite the laser induction being on the membrane surface.

The laser-induced membrane discussed above was prepared with a laser fluence of 21.2 mJ cm$^{-2}$. To elucidate the differences in electrochemical performance under varying laser energies, tests were conducted on LPAc membranes treated with laser fluences of 15.8 mJ cm$^{-2}$ and 28.3 mJ cm$^{-2}$. The membrane surface treated by different laser energies is shown in Fig. S7a. The polarization curves and power density curves of PBI and LPAc with different laser fluences are shown in Fig. S7b. As shown in Fig. S7b, the impact of lower laser energy on the PBI is minimal, whereas higher laser energy can lead to hydrogen crossover, which is reflected in the lower OCV. This, in turn, affects the performance improvement. Therefore, precise control of laser energy is crucial for optimizing performance enhancement.

**Morphology and pore analysis of the MEA**

To visualize the distribution in composition of the different MEAs after ASTs, and help further understand the differences in performance and durability, μ-CT was employed. Volume renderings of segmented PBI, LPAa, LPAc, and LPAac are all shown in Fig. 3a. The segmented volume renderings of La, Lc, and Lac are shown in Fig. S8a. Typical segmented orthoslices from different MEAs used for analysis are shown in Fig. S9. While lab-based μ-CT has numerous advantages, its ability to distinguish between nano-micro scale objects with low atomic mass is somewhat limited. Hence, components in the MPL and GDL could only be adequately segmented into two phases: a pore phase and a mixture phase. The mixed phase is defined as the phase within the Gas Diffusion Electrode (GDE) that is not detected as pores by μ-CT, which include nano-pores not detectable by micro-CT; Materials within the mixture phase include leached PA, PTFE binder, carbon fibers, carbon

powder, and any water present. Despite this limitation, notable differences in porosity among different MEAs could still be observed, allowing for a comparison of the migration and redistribution of different MEA components. The radar chart in Fig. 3b illustrates the average mixture phase volume fraction for PBI, LPAa, LPAc and LPAac. The equivalent chart of La, Lc and Lac is shown in Fig. S8b. The mixture phase volume fraction is defined as the volume of the mixture phase divided by the sum of the volumes of the mixture phase and the pore phase. The average mixture volume fractions for both the anode and cathode sides of PBI can be seen to exhibit minimal differences, indicating relatively balanced leaching of PA on both sides. In contrast, for LPAa and LPAc the leaching of PA on the side without laser treatment is higher than on the laser-treated side, a contrast is significantly most pronounced in the case of LPAa, suggesting significant PA leaching on the cathode side. The polarization curve of La shown in Fig. S2 exhibits a sudden drop in polarization after AST at intermediate current densities, also reflecting the reaction induced by laser treatment on the anode side. The instantaneous accumulation of a large amount of PA and water on the cathode side leads to catalyst flooding, thereby instantaneously obstructing mass transfer. Despite a slight difference in the mixture phase volume fractions between the two sides of LPAac, its mixture component on both sides is the lowest among all MEAs, implying little PA leaching occurred. This further suggests that the laser treatment of the PBI has a significant influence on membrane behavior.

Due to the strong shielding effect of high-loading Pt-based metal catalysts on X-rays, while μ-CT cannot investigate the porous structure within the CL, it can provide a clear overall morphology of the CL due to the high concentration of Pt. From the cross-sectional views of the volume rendering in Fig. 3a and Fig. S8a, and orthoslices in Fig. S9, MEAs treated with the laser exhibited a more pronounced sawtooth distribution of catalyst compared to PBI. This is attributed to gaps in the laser footprint, resulting in localized variations in the laser-induced graphene on the membrane surface and influencing PA leaching differently. The leaching of PA could, in turn, drive catalyst migration. This is likely attributed to the erosion caused by PA fluid flow on the CL. Despite laser treatment being applied only to one side of the membrane for LPAa and LPAc, their post-AST anode and cathode sawtooth catalyst distributions display a certain degree of symmetry. This suggests mutual influences between anodic and cathodic PA or catalyst migration (see below). This sawtooth distribution could potentially facilitate the optimization of the three-phase interface of pre-AST LPAa and LPAc are shown in Fig. S8a, b, respectively. It can be observed that the CL of pristine LPAa and LPAc does not exhibit a sawtooth-like distribution (shown in Fig. S10), indicating that this distribution occurs under the influence of electrochemical actions in AST. Taking PBI and LPAac as examples for comparison, a mechanism of the laser-induced effects on the three-phase interface can be proposed, as illustrated in Fig. 3c. For HT-PEMFC based on PA doped PBI membranes, PA leaching occurs during the AST process, primarily due to the dominance of anodic electrochemical pumping and cathodic water production. Moderate PA leaching is advantageous for establishing proton transport pathways between the catalyst and membrane, activating the HT-PEMFC. However, excessive PA leaching can lead to the isolation of gas from the catalyst, affecting the three-phase interface and causing a gradual decline in performance. The graphene barrier formed by laser-induced treatment on the surface of the PBI membrane can globally slow down PA leaching, as demonstrated by optical microscopy in Fig. 1g, h. This is further illustrated in Fig. S11, which shows the time dependence of the weight loss ratio of acid. However, PA can still leach from the gaps in the laser footprint, leading to a sustainable degree of catalyst migration. This localized difference can cause a sawtooth distribution of the electrolyte in the catalyst layer, influencing the three-phase interface and allowing more catalysts to be

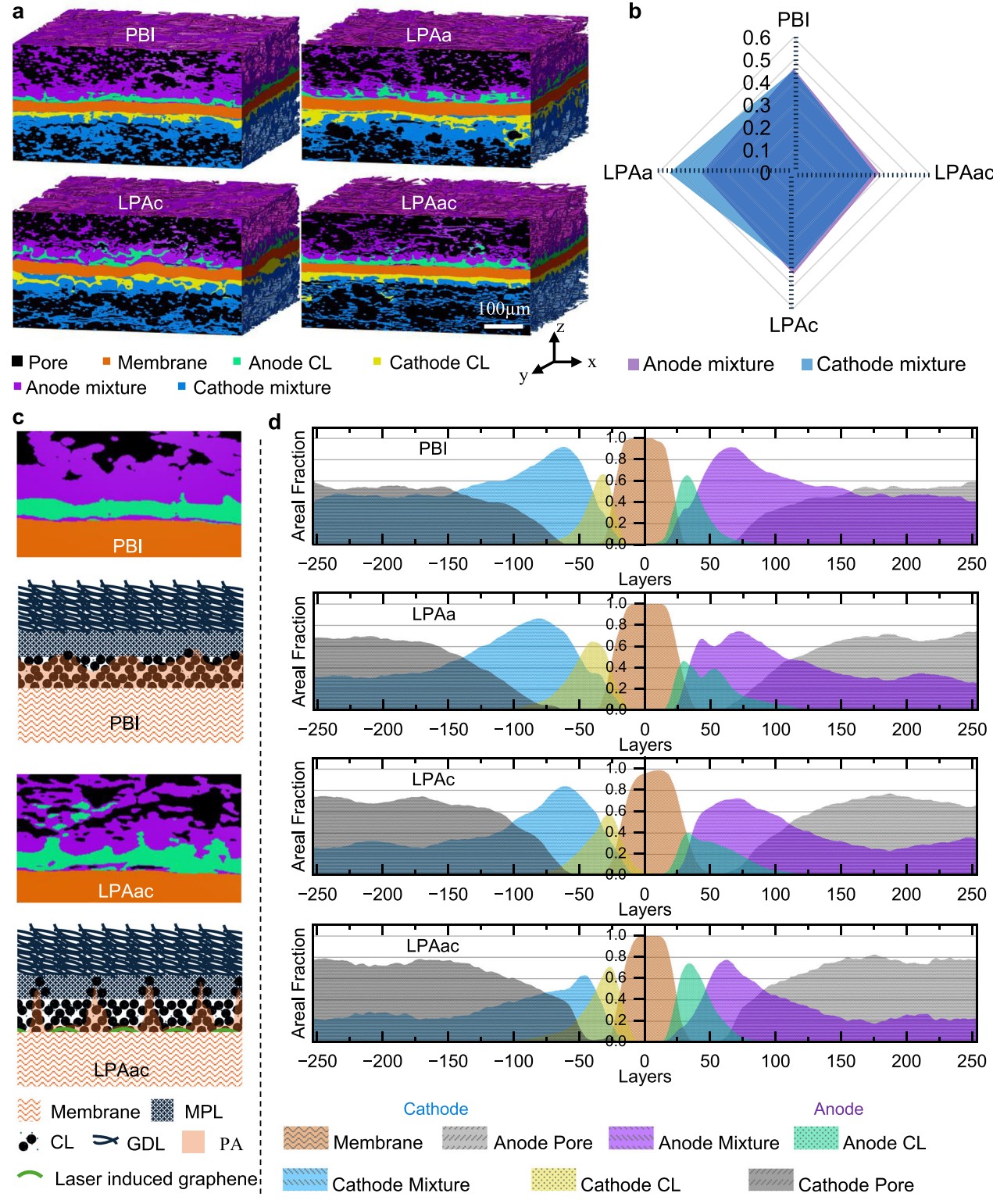

**Fig. 3 | The three-dimensional structure and component spatial distribution of MEA. a** Volume rendering of segmented MEAs. **b** Electrode average mixture phase volume fraction. **c** Schematic illustration of catalyst migration and the three-phase interface mechanism. **d** Slice-by-slice plots of average area fraction of volume rendering in Z direction.

effective in reducing charge transfer resistance and increasing proton transport channels.

The slice-by-slice average area fraction plots in Fig. 3d and Fig. S8c provide a more intuitive visualization of component distribution and material migration in the different MEAs. Comparing PBI and LPAac,

the peak of the mixture phase volume fraction curve for PBI is more inclined towards the outer edges of the MEA, and the mixture phase volume fraction component of PBI is relatively larger, extending from the side near the membrane to the side adjacent to the gas channel side. Additionally, the CL peaks for both PBI and LPAac are relatively

convergent, which indicates a relatively weak migration of the catalyst. While, the CL peaks for LPAa and LPAc, both induced by laser treatment on one side, are relatively divergent indicating a wider distribution range of CL across the cross-section, especially on the anode side. This is likely related to the previously mentioned sawtooth distribution.

Among the different MEAs, in addition to the notable differences in CL morphology, there are apparent variations in membrane morphology after AST, reflected in the curvature of the membrane surface. The curvature of the membrane surface refers to the degree of deviation from being flat or planar, describing the extent to which the membrane surface is curved or bent in three-dimensional space. It quantifies the deviation of the surface from a straight line or a plane at any given point. The laser induction on the membrane surface, especially when applied to one side of the membrane, result in various changes in the membrane, including water absorption, PA leaching, mechanical properties, and heat transfer. These changes could lead to differences in membrane curvature. Greater membrane curvature indicates a more convoluted membrane surface, which, under the same active area of the electrode, results in a larger three-phase interface. This consequently enhances the effective utilization of the catalyst and provides additional pathways for proton transfer.

Stress-strain curves for pure PBI and PA-doped PBI membranes subjected to single-sided and double-sided laser induction are shown in Fig. S13. The linear regions of the different curves coincide, indicating that the Young's modulus remains relatively unchanged. However, laser induction significantly reduces the yield strength. Notably, the tensile strength of membranes subjected to single-sided laser induction decreases, making them more prone to deformation. This phenomenon is attributed to the differences between the two sides of the membrane and the uneven stress distribution caused by the roughness of the single-treated surface. In contrast, membranes subjected to double-sided laser induction exhibit an increase in tensile strength. Additionally, both single-sided and double-sided laser-induced membranes show an improvement in elongation at break. The impact of laser induction on the mechanical properties of PA-doped PBI membranes could result in varying degrees of deformation within the MEA. The membrane surface curvature distributions of PBI, LPAa, and LPAc are illustrated in Fig. 4a and surface curvature distributions of La, Lc, Lac and LPAac are illustrated in Fig. S12a. It is evident that the surface irregularity of LPAa and LPAc membranes is significantly higher than that of PBI. However, there is little difference between La, Lac, LPAac, and PBI. This can be more clearly contrasted in the statistical plot shown in Fig. 4b. The membrane with the highest curvature is LPAc, which also exhibits the best performance. Although both LPAa and LPAc membranes were subjected to unilateral laser treatment, several differences exist between the cathode and anode in HT-PEMFCs. The cathode involves water generation, whereas the anode does not. Additionally, the mechanisms of PA leaching differ between the anode and cathode. Different laser treatments can lead to varying degrees of PA leaching, and the PA content affects mechanical properties. The combined thermal effects and compositional migration on both sides contribute to the varied curvature observed in LPAa and LPAc membranes.

Although an increase in membrane curvature could result in gaps between the membrane and the catalyst layer, the CT images in Fig. 3a show that any voids formed between the LPAc membrane, and the catalyst layer are in mixture regions. Combined with the electrochemical characterization indicating superior electrochemical performance for LPAc, this suggests that the PA extruded due to membrane bending fills the gaps between the membrane and CL. These gaps between the membrane and the catalyst layer also provide a buffer for PA to inundate the CL. In the case of LPAac, which has been induced by laser on both sides, the membrane does not undergo significant bending deformation, reducing the occurrence of PA leaching due to

compression, which suggest that the bending is driven by a difference in the tension between the two sides of the membrane which enables deformation. As the laser-induced graphene can impede the leaching of PA, this also contribute to the lower mixture fraction observed for LPAac. The equivalent curvature statistical plots of La, Lc and Lac are shown in Fig. S14a. Further exploration can be conducted by considering the pore and isolated pore distribution in Fig. 4c–e. The curvature of different pristine MEA membranes before AST is illustrated in Fig. S14b. It can be observed that after hot pressing, the curvature of different pristine MEA membranes is relatively small and does not differ significantly. This indicates that the change in membrane curvature is also caused by the electrochemical processes accompanying AST.

The schematic representation of the ball-and-stick model for the pore and throat network of the electrode in the MEA based on the PBI membrane, along with the distribution of isolated pores, is illustrated in Fig. 4c. The pore and throat network of the electrode refers to the interconnected system of void spaces (pores) and narrow channels (throats) within the electrode structure. These features facilitate the transport of reactants and products during electrochemical reactions within the fuel cell electrode. Isolated pores in fuel cell electrodes are individual void spaces within the electrode structure that are not connected to the surrounding pore network, potentially hindering efficient mass transport. From the comparative plot of the average number of pores and throats of PBI, LPAa, LPAc and LPAac shown in Fig. 4d (the equivalent plot for La, Lc and Lac are shown in Fig. S14c), the differences in pore and throat numbers correspond closely to the relative porosities depicted in Fig. 3a, b. However, the variations in the distribution of isolated pores shown in Fig. 4e have a more significant impact on the performance of the MEA. Despite the substantial PA leaching in PBI, this leached PA fills the pores adjacent to the membrane on the catalyst side. Therefore, PBI has fewer isolated pores. In contrast, LPAac has relatively flat membranes, and both sides of the membrane are blocked by laser-induced graphene material, preventing PA leaching. This leads to gaps between the membrane and the CL, without electrolyte filling. As a result, the distribution plot in Fig. 4e shows a significant number of isolated pores near the membrane. This can also be observed in the CT images in Fig. 3a and Fig. S8a. The slice-by-slice plots of isolated pores of La, Lc and Lac are illustrated in Fig. S14d the relevant discussion is in supporting information.

Based on the segmentation of CT scan structures and pore analysis, it is feasible to determine the permeability of hydrogen and oxygen in GDL driven by pressure, without the involvement of electrochemical reactions. Using PBI's GDL as an example, the simulation results are depicted in Fig. 4f. The statistical plots of average absolute permeability for PBI, LPAa, LPAc and LPAac are shown in Fig. 4g (La, Lc and Lac in Fig. S14e). From the simulation results, it is evident that a higher number of pores and throats does not necessarily indicate better permeability. Permeability analysis can better illustrate the interconnected structure of pores and the pore network under the influence of isolated pores. Permeability analysis can provide essential parameters for the subsequent visualization-based simulation of MEA based on electrochemical reactions.

## Multi-physics electrochemical simulations based on the three-dimensional scanned structure of MEAs

μ-CT enables the visualization of the three-dimensional structure of MEAs. To further achieve the visualization of electrochemical processes within MEAs, coupled simulations of electrochemical performance and free porous media flow were conducted in COMSOL. Due to the superior performance of LPAa and LPAc after AST, we focused on simulating and comparing the differences in the electrochemical processes within MEAs containing LPAa, LPAc, and LPAac membranes. The simulation used air instead of pure oxygen to more clearly

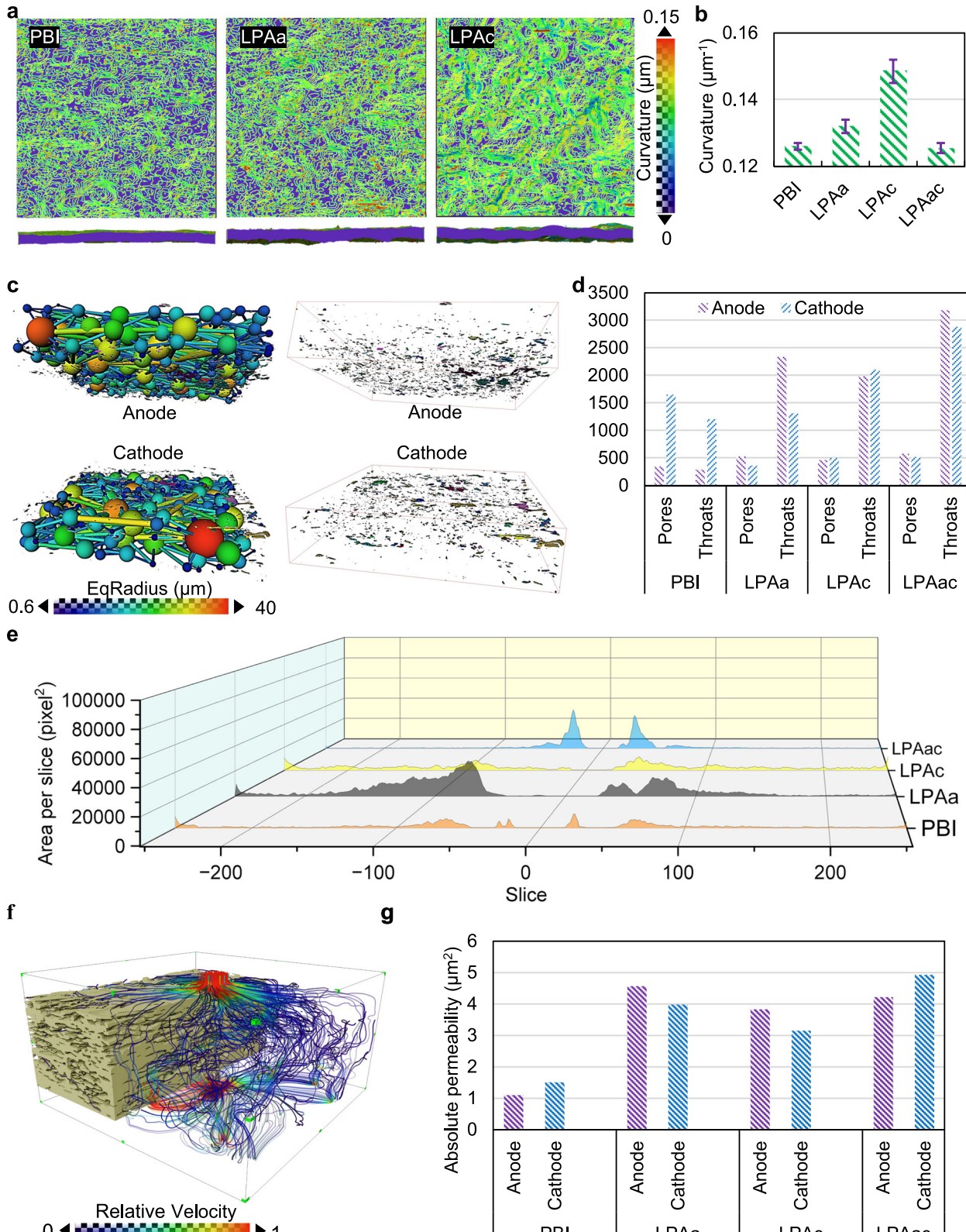

**Fig. 4 | Pore structure and spatial distribution in MEAs after AST. a** The curvature distribution plots of the membrane. The upper portion consists of surfaces characterized by iso-curvature lines, while the lower portion depicts interface diagrams. Transitioning from purple to red signifies an increase in curvature. **b** The statistical distribution plot of the average curvature of the membrane. The error bars reflect the variability across different membrane regions and between samples. **c** The ball-and-stick model of the electrode pores and throat network, along with the distribution plot of isolated pores. **d** Statistical plots depicting the average number of pores and throats in electrodes across different MEAs. **e** Slice-by-slice plots of isolated pores of different MEAs. **f** illuminated streamlines of absolute permeability in the pore area. **g** statistical plots of absolute permeability for different MEAs' pore regions.

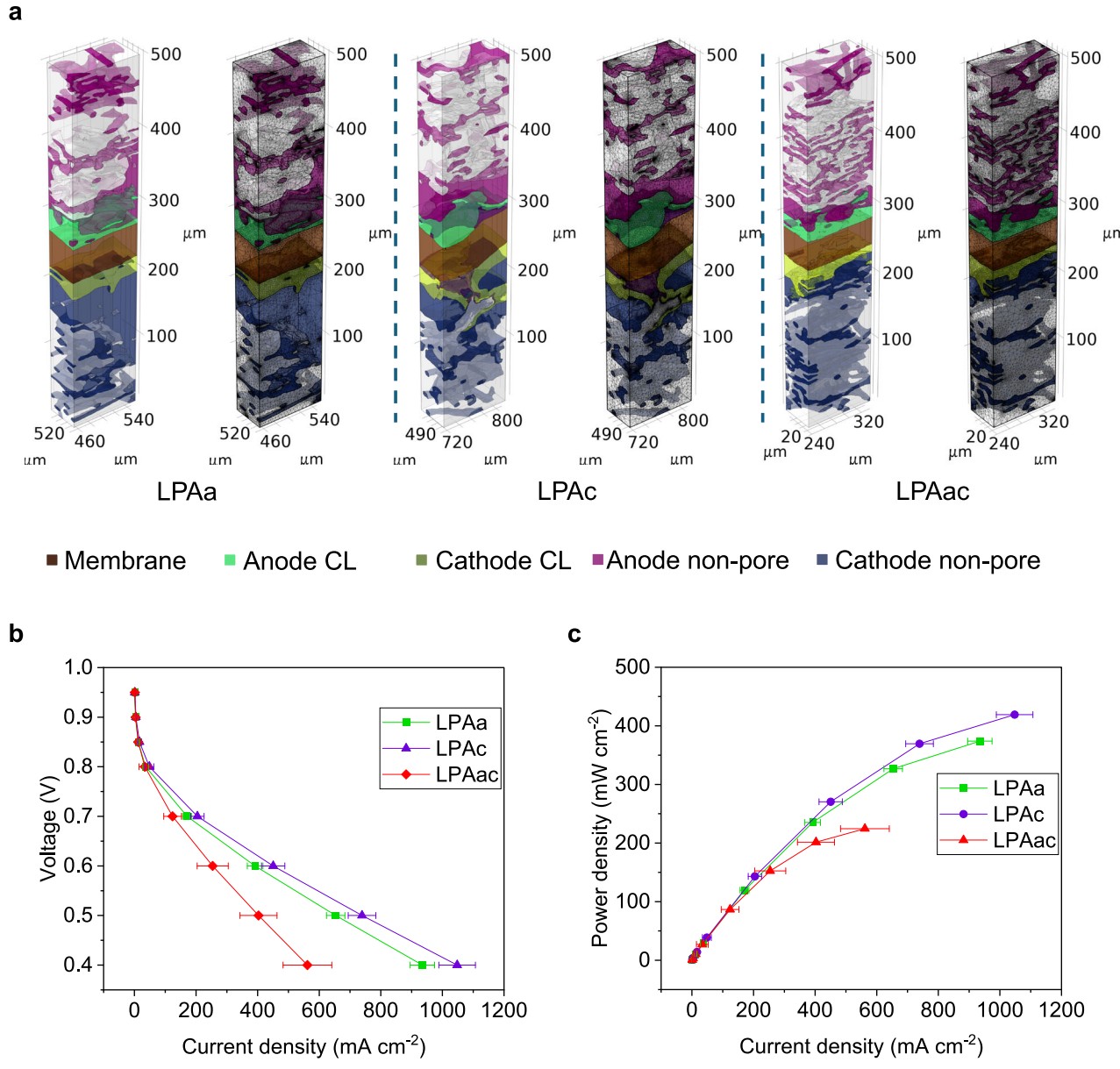

**Fig. 5 | 3D model and electrochemical performance of multi-physics coupling visual simulation. a** The typical MEA structures and their meshing used for COMSOL simulation of LPAa, LPAc, and LPAac. **b** The polarization curves obtained from simulations. The error bars reflect the variability across different models. **c** The power density curves obtained from simulations.

illustrate the impact of material migration on gas distribution. By reducing the mole fraction of oxygen at the cathode, the effects of membrane deformation, catalyst migration, and PA transport on mass transfer are highlighted, showing a performance decline under caustic conditions, different to the oxygen used in the experiments. The typical MEA structures used for the simulations of LPAa, LPAc, and LPAac are shown in Fig. 5a, while Fig. S15 illustrates additional 3D MEA structure examples reconstructed through CT scans for simulation purposes. As the simulations were conducted using finite element analysis, the three-dimensional structure of the MEA was meshed, this is also illustrated in Fig. 5a. The parameters utilized in the multi-physics simulation were derived from both the electrochemical characterization performed and parameters from CT data processing. As they were below the resolution limit for μ-CT, it was not possible to determine the exact pore fraction and permeability of the micro-porous structure in the CL.

Based on the parameter settings reported in previous literature for simplified 2D or 3D models of HT-PEMFCs[33–35,63,64], it is assumed that the permeability ratio between the nanopores in the CL or mixture phase and the micron-scale pores in the GDL remains consistent. Focused Ion Beam Scanning Electron Microscopy (FIB-SEM) was employed to investigate the three-dimensional spatial distribution of nanopores within the CL and to conduct permeability simulations. The FIB-SEM scanning results and permeability simulation diagrams are shown in Fig. S16a, b. By calculating the average permeability in the CL and GDL, this ratio was determined to be 6.25. However, due to the impracticality of extending nanoscale scanning to all components of the entire MEA, the CL was modelled as a porous homogenous layer. This simplification is one of the primary sources of error in this simulation approach. By applying a domain probe to the modelled anode CL, to obtain the current density at different voltages

(0.95 V–0.4 V), the polarization curves and power density curves shown in Fig. 5b, c were obtained. The simulation and experiment maintained consistency in parameters such as temperature, and stoichiometric ratios of hydrogen and oxygen, but to better reflect the influence of the MEA structure on material transport, compressed air was used on the cathode side during simulation instead of pure oxygen. Despite the partial disparity observed in the polarization curves obtained from the simulation and experiment, the performance comparison among LPAa, LPAc, and LPAac remains consistent between experiments and simulations. In addition to the aforementioned factors, discrepancies between simulation and actual conditions can also arise from the precision of the scanning process, the accuracy of segmentation, and the fidelity of finite element meshing.

As the simulation is based on the Multiphysics coupling of free and porous media flow, numerous processes within the MEA can be visualised; The electrochemical reactions under a specific MEA morphology, the streamlining of gas transport in the GDE, the distribution of voltage in the electrolyte and electrode at different output voltages, and the vector distribution of current density, can be simultaneously extracted. Select simulation results, where the results for LPAa are presented as an example, are shown in Fig. 6. The dynamic results of the simulated outcomes of LPAa, as exemplified by the Supplementary Movie 1, exhibit the variation of results with changing voltage. Total flux streamline of $H_2$, water mole fraction distribution, electrolyte potential distribution, total flux streamline of $O_2$ and electrode potential distribution with respect to the ground of different MEAs at different voltages are shown in Fig. S17 to Fig. S25.

The data in Fig. 6 shows how the cathodic oxygen streamline, potential distribution in the electrode and electrolyte, and current density within LPAa change with increasing voltage polarisation. Based on the previous CT data, it is evident that the cathode of LPAa has the highest mixture phase fraction, representing a relatively significant level of PA leaching. Fig. 6a illustrates that with increasing voltage drop and current density, extensive regions of the mixture phase become a crucial factor in reducing the oxygen molar fraction at the corresponding catalyst location. Comparing the membrane component distribution in Fig. 5a with the multislice of electrolyte potential distribution in Fig. 6b, it is evident that the simulated electrolyte is not confined solely to the membrane region. Instead, it integrates the distribution in conjunction with the specific MEA structure, accounting for the impact of electrochemical processes on the distribution of PA leaching. The arrow volume, representing the current density vector, also illustrates the process of proton transfer. Due to the notable difference in PA leaching between the anode and cathode within LPAa, a distinct distribution of the mixture phase is established. This disparity also results in a clear contrast in the current density distribution between the two sides of the membrane. If larger regions of the mixture phase were present, it is logical from these results that this would facilitate a more uniform distribution of electrode potential at low voltages, i.e., at high current densities.

The convergence area combines gas distribution, the potential distribution of electrolytes and electrodes, gas flow velocity, as well as current density distribution. These elements happen to effectively construct the three-dimensional three-phase interface of fuel cells. Local mass transfer, momentum transfer, and electrochemical reaction information can be obtained for points in a three-dimensional coordinate system. These pieces of information can effectively reflect the impact of material migration and structural morphological changes on performance and durability in HT-PEMFC, thus facilitating diagnosis and optimization. Additionally, coupling under multiple electrochemical parameter variables can be achieved within a three-dimensional framework.

Visual simulation results of different MEAs under identical conditions can effectively explore the impact of varying MEA structures and material migration on mass transfer, momentum transfer, and electrochemical reactions. Fig. 7 illustrates the hydrogen and oxygen transport characteristics of LPAa, LPAc, and LPAac at 0.4 V. At 0.4 V, the high current density stage of MEAs better reflects the mass transfer under high consumption of hydrogen and oxygen. Based on the distribution of hydrogen molar fractions shown in Fig. 7a, it can be observed that although the hydrogen transport to the CL is slightly affected by PA leaching in LPAa and LPAc compared to LPAac, the molar fraction of hydrogen in the anode CL region is relatively almost identical across all of the different MEAs (~0.963). It is therefore evident that the mass transfer resistance at the anode is not the primary factor influencing the performance of the MEA. Due to the more complex electron transfer processes involved in the oxygen reduction reaction (ORR) at the cathode compared to the hydrogen oxidation reaction (HOR) at the anode, the transport of oxygen in the MEA is more critical[65,66]. The molar fraction of oxygen in the cathode CL region varies significantly among different MEAs, as observed in Fig. 7a, b. LPAa notably exhibits the poorest oxygen transfer. This difference is further reflected in the distribution of oxygen velocity, as shown in Fig. 7c. While the oxygen velocity distribution is similar for LPAc and LPAac, LPAa shows a concentration of oxygen on the side closer to the gas channel, with significantly lower gas velocity on the CL side in the GDE. This indicates that for LPAa, oxygen rapidly flows near the GDE on the cathode side closer to the gas channel without effectively permeating towards the catalyst side. The higher mass transfer resistance observed in LPAa compared to LPAc contributes to its relatively poorer performance. However, despite exhibiting the best oxygen and hydrogen flux in the CL region, LPAac demonstrates performance inferior to LPAa and LPAc in both experimental and simulation results.

Figure 8 illustrates the distribution of voltage, current density, and water mole fraction in MEAs containing LPAa, LPAc, and LPAac membranes. From Fig. 8a, it can be observed that LPAa and LPAc exhibit a more varied distribution of electrolyte potential compared to LPAac. Although this represents a larger voltage drop across the electrolyte to some extent, the local differences in electrolyte potential are more significant at the boundaries for LPAa and LPAc compared to LPAac. This indicates that in comparison to LPAac, the electrolyte in LPAa and LPAc diffuses more effectively towards the catalyst, acting similarly to the ionomer, especially in the case of LPAc. The boundary of the electrolyte in LPAa and LPAc appears more saw-toothed compared to LPAac, enhancing the three-phase interface. This is also reflected in the more extensive and varied distribution of current density vectors for LPAa and LPAc compared to LPAac, where effective diffusion of electrolyte in the CL increases proton transfer channels and utilization of the catalyst. This inference can be supported by considering the electrode potential and current density vector distribution, as well as the water mole fraction distribution shown in Fig. 8b, c. Compared to LPAac, LPAa and LPAc exhibit a more uniformly dense current density vector in the CL region. It is noteworthy that the larger mixture phase at the cathode of LPAa, while potentially causing mass transfer resistance, contributes to a more uniform distribution of current density vectors in the electrode. In the case of LPAc, the anode CL, possibly due to an optimal level of PA leaching, shows a more uniform distribution of current density vectors compared to LPAa and LPAac. Therefore this demonstrates that, in addition to the membrane's intrinsic conductivity, the efficient transition of potential and current density from the electrolyte to the electrode is crucial for optimal performance. The water Mole fraction distribution in Fig. 8c provides insights into the rate of cathodic electrochemical reactions. The noticeably higher water content at the

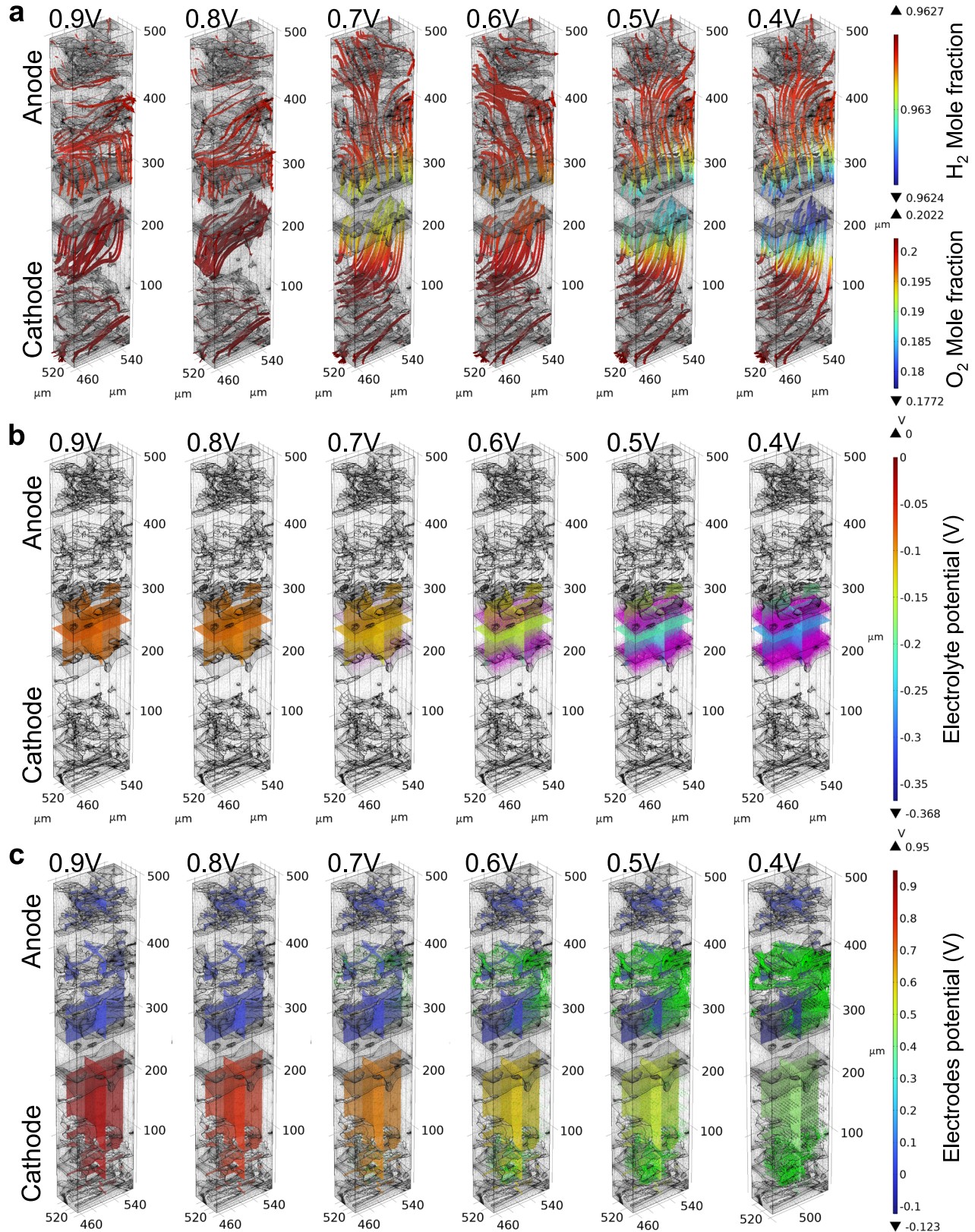

**Fig. 6 | Images depicting the simulation results of LPAa varying with voltage. a** Total flux streamline of O$_2$. **b** Electrolyte potential (arrow volume: electrolyte current density vector), **c** electrode potential with respect to ground (arrow volume: electrode current density vector).

cathode in LPAa and LPAc, compared to LPAac, suggests higher reaction rates. However, it cannot be ruled out that the poorer mass transfer in LPAa and LPAc contributes to water accumulation. LPAa could be more constrained by water accumulation-induced material transfer resistance, while LPAc exhibits superior water content, facilitating the efficient transfer of protons and electrons at the three-phase interface, thereby demonstrating the best performance.

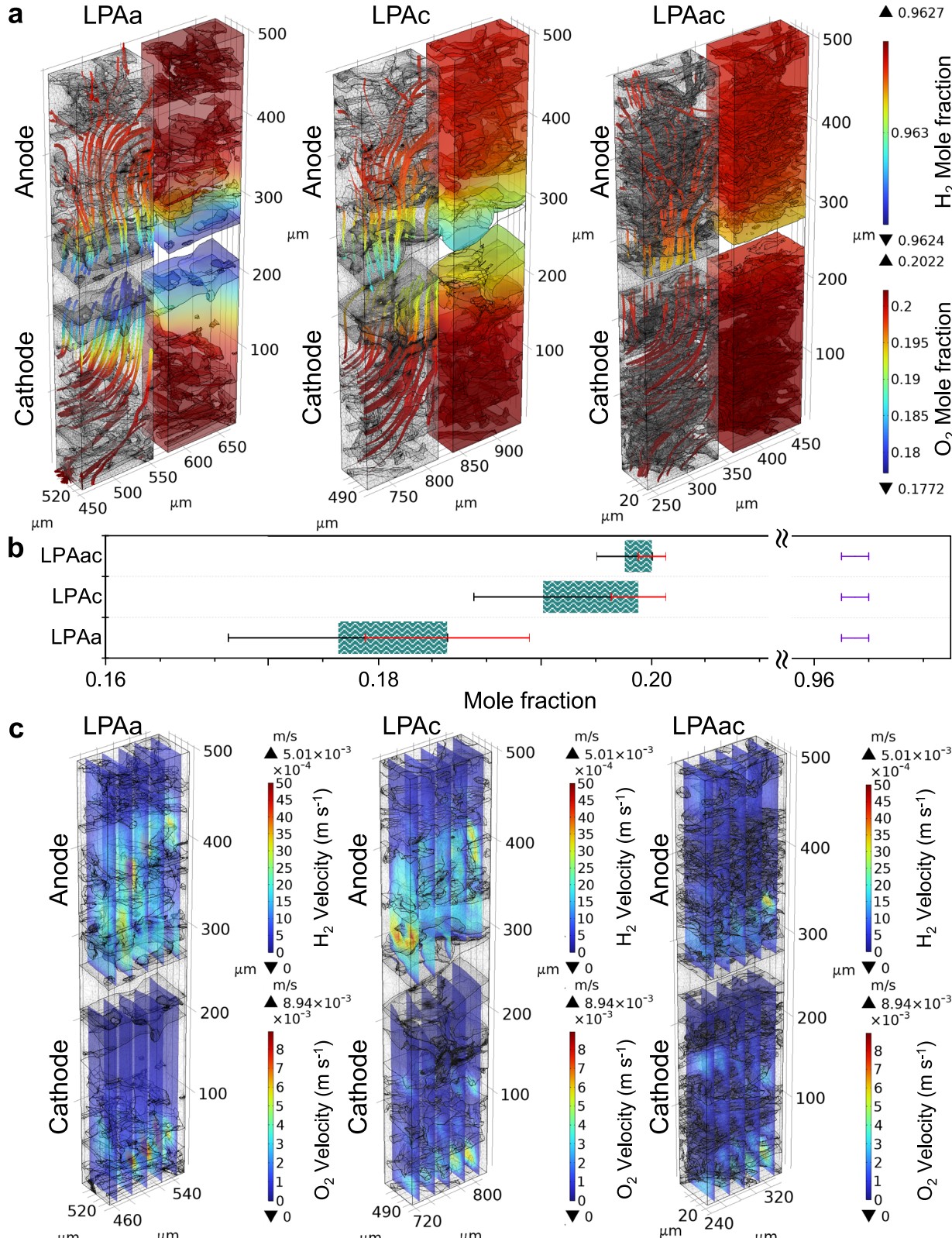

**Fig. 7 | Visualization simulation and statistics of gas distribution and velocity at 0.4 V. a** Total flux mole fraction surface and streamline of H$_2$ and O$_2$. **b** Statistical distribution of the molar fractions of oxygen and hydrogen in the catalyst layer. The green area and purple bars represent the distribution range of the molar fraction of oxygen and hydrogen, respectively. The black and red bars correspond to the error bars indicating the minimum and maximum values of the molar fraction of oxygen, respectively. **c** velocity magnitude slices of O$_2$.

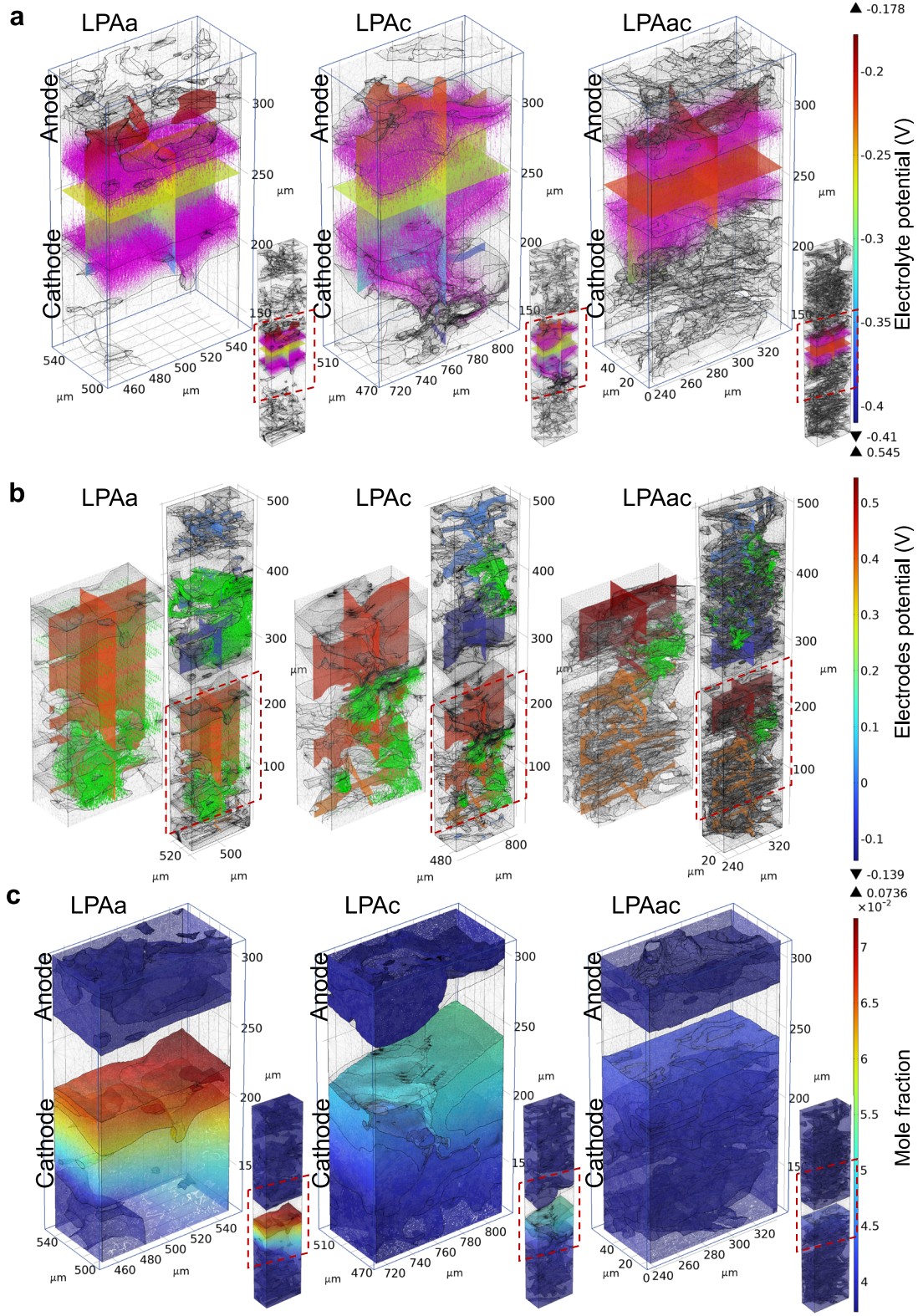

**Fig. 8 | Visualization simulation of potential, current density and water distribution at 0.4 V. a** Electrolyte potential (arrow volume: electrolyte current density vector). **b** electrode potential with respect to ground (arrow volume: electrode current density vector). **c** water mole fraction distribution surface.

Overall, this integrated visualization simulation combining multiple physical fields and phases provides a clear interpretation of the impact of the three-phase interface on electrochemical reactions in a real MEA structure. This knowledge will prove crucial for optimizing MEA structures.

Here, through grid-patterned picosecond laser induction on the surface of PBI membranes before and after PA doping, a membrane interface capable of controlling PA leaching was created. This method is quick and scalable, forming a foam-like two-dimensional graphene structure on the membrane surface, accompanied by a degree of

oxidation. By comparing MEAs prepared using a variety of conditions it was revealed that laser induction could effectively enhance the performance and durability of HT-PEMFC, especially in the case of laser induction on the cathode side after PA doping. MEAs based on this membrane structure achieved a 58.2% increase in peak power density after AST compared to an untreated PBI membrane due to lower charge transfer and mass transfer resistances.

The application of X-ray μ-CT enables the three-dimensional visualisation of these MEA structures, allowing the development of an image-based multi-physics HT-PEMFC MEA model to uncover the influence of MEA materials and methods of manufacture, including picosecond laser induction, on cell performance; Distribution of electrolyte and electrode potentials, current density vectors, mole fractions of hydrogen, oxygen, and water, as well as oxygen velocity distribution, could all be extracted. The dynamics of these parameters indicated that the performance of HT-PEMFC is not solely dependent on the individual dynamics of each component but is instead influenced more by the interplay of different phases. The localized foam-like graphene oxide structures formed by laser treatment can affect the penetration of PA and result in a sawtooth distribution of PA within the catalyst layer, thereby increasing the triple-phase boundary. Additionally, the asymmetry between the two sides of the membrane caused by one-sided laser treatment can lead to increased membrane curvature, further enhancing the triple-phase boundary. Moreover, the catalyst migration resulting from this effect, along with the differences in the pore network of the GDL, further contributes to the variations in the triple-phase boundary. Specifically, here, this model showed that the MEA treated with laser induction on the cathode side after PA doping exhibited enhanced performance due to an optimized three-phase interface and unhindered charge or ion conduction between components. More widely, this methodology offers an effective method to understand the development of complex buried interfaces within MEAs without the use of destructive or ex-situ characterization tools, with potential impacts for a wide range of technologies that rely on membrane-electrode systems.

## Methods

### Materials
Carbon paper (TGPH 090, Toray) infused with 5 wt% polytetra-fluoroethylene (PTFE), provided essential electrical conductivity and structural support for the HT-PEMFC MEA. The microporous layer (MPL) ink used in this study was composed of 90 wt% Ketjen Black (EC-300J, AkzoNobel) and 10 wt% PTFE (Sigma Aldrich) dispersed in isopropanol. The catalyst utilized on the anode and cathode was Pt/C (60% Pt on high surface area advanced carbon support, HiSPEC™ 9100, Fisher Scientific). The PBI membrane (Fumapem® AP-30, Fumatech) contained orthophosphoric acid (85 + %, Extra pure, Fisher Scientific), with an 85 wt% concentration.

### Preparation of LIG/PBI Membrane
The PBI membranes and PA-doped PBI membranes were positioned on a platform for exposure to a UV picosecond pulsed laser (Edge wave 100 W) with a 355 nm wavelength and a 10 ps pulse duration. The laser beam was directed at the target surface with a 40 μm spot size. Utilizing a Galvo head for precise control, the laser fluences was 15.8 mJ cm$^{-2}$, 21.2 mJ cm$^{-2}$ and 28.3 mJ cm$^{-2}$. The pulse repetition rate was consistently set at 3372.8 kHz, with a scan speed of 200 mm s$^{-1}$, and a 20 μm spacing between the scanning lines.

### Membrane characterization
The surface morphology of the membrane was examined using a Tescan Mira3 SC SEM operating at 5.0 kV in secondary electron mode. Energy dispersive X-ray spectroscopy (EDS) analysis was conducted using the EDS attachment from Oxford Instruments on the same Tescan Mira3 SC instrument. Raman spectroscopy was carried out

utilizing a Horiba LabRAM Evolution equipped with a 488 nm laser, and the Raman shift was recorded in the range of 1000 cm$^{-1}$ to 3000 cm$^{-1}$. The laser-induced membrane and the laser-induced membrane with PA doping were also examined using high-resolution optical microscopy on a Keyence VHX-7000 optical microscope (UK) equipped with a 400× objective lens. The acid retention test involved suspending the PA-doped membranes above boiling water for 5 h[67]. The weight of the membranes was recorded every hour to measure the amount of acid leached. Stress-strain test was performed utilizing a Q800 Dynamic Mechanical Analyzer (DMA) in controlled force mode. Grazing–incidence X–ray diffraction (GIXRD) data were collected by Malvern Panalytical Empyrean X–ray diffractometer. The radiation source was Cu Kα (λ = 1.5406 Å) operated at 40 kV, 40 mA with an incidence angle ω = 1°. All patterns were obtained by using a step scan method (0.05° per step for 1 s), in a 2θ range from 5° to 60°.

### Preparation of MEA
The MPL ink and CL ink were coated sequentially onto carbon paper through cross-spraying. Subsequently, the coated electrodes were heated on a hot plate at 120 °C to eliminate the solvent until a loading of 1 mg cm$^{-2}$ was reached for both Ketjen black and Pt. The PBI membrane underwent submerging in PA at 140°C with an oil bath for four hours to get the PA-doped membrane. The electrodes and membrane were hot-pressed at 80 psi at 140 °C for 4 min. 150 μm-thick PTFE films were employed as gaskets in the final assembly of the MEA.

### Electrochemical methods and the characterization of membranes
The MEA was mounted in a Scribner fuel cell fixture with an effective area of 5 cm² and subjected to a pressure of 5 N·m. Performance testing of the MEAs was conducted under conditions where oxygen and hydrogen were fed to the cathode and anode, respectively, at stoichiometric ratios of 2 and 1.2, at a temperature of 160 °C. A GAMRY potentiostat (Reference 3000 with 30 k booster) was utilized to conduct measurements, including polarization curves, electrochemical impedance spectroscopy (EIS), and accelerated stress tests (AST). A series of polarization curves were generated by discharging from open circuit voltage (OCV) to 0.1 V in increments of 0.1 A with a 5 s dwell time. Galvanostatic EIS was performed at a current density of 0.5 A cm$^{-2}$, spanning a frequency range from 10 kHz to 0.1 Hz, to obtain Nyquist plots. The MEAs were subjected to an AST protocol, which involved repeated chronopotentiometry with a 4-minute operation at 0.6 A cm$^{-2}$ followed by a 16-min operation at 1.0 A cm$^{-2}$. The cells were held at OCV for 10 min between each cycle. This cycle was repeated to total 6 h. The LSV measurements were conducted using a GAMRY Reference 3000. During the LSV measurement, the voltage was increased from 0 to 550 mV at a scan rate of 1 mV s$^{-1}$ and a step size of 2 mV. The proton conductivity is calculated by the equation:

$$\sigma = \frac{L}{R \times A} \tag{1}$$

σ represents the proton conductivity (S cm$^{-1}$), L denotes the membrane thickness (cm), R stands for the membrane's internal resistance (Ω), and A signifies the membrane's active area (cm²).

### X-ray micro-CT
Circular discs with diameters of 2 mm were milled from the active area using a laser micro-machining tool (Oxford Lasers, A Series/Compact System) operating at approximately 0.6 W and a scan speed of around 0.5 mm s$^{-1}$, with 5 iterations. A Carl Zeiss, Zeiss Xradia 620 Versa X-ray micro-CT instrument was employed for CT scans, equipped with a polychromatic source with a tube voltage range of 60 kV. A 4× objective was selected to achieve a voxel dimension of 1 μm, providing

a field-of-view of about 2 mm at suitable source-to-sample and sample-to-detector distances. The exposure time was set at 8 s.

## Data analysis and modelling

The CT data obtained through scanning was cropped to a size of $1016 \times 1016 \times 508$ pixels in Avizo to reduce computational workload. The size of each voxel was 1 μm. Following the cropping, a non-local means filter was applied to enhance the image clarity. The MEA was subjected to deep learning using AI algorithms for segmentation to distinguish the membrane from other components. The remaining components were segmented in Avizo into the catalyst layer, pores, and the mixed phase. The mixed phase includes carbon particles, carbon fibers, PTFE binder, and PA extracted from the membrane in both the MPL and GDL. Based on the segmented MEA structure, porosity and connectivity analyses were conducted in Avizo to establish the distribution of isolated pores and the pore network model. The CT data was further cropped to a size of $100 \times 100 \times 50$ pixels for multi-physics coupling simulation in COMSOL to minimise the computational time. Simulation in COMSOL is accomplished by observing the mass and momentum transport phenomena of reactants, and electrochemical currents under different MEAs structures in steady-state. Additionally, simulations in COMSOL involve interfaces related to hydrogen fuel cell interface, free and porous media flow interfaces, and Multiphysics nodes encompassing reactant gas phase and reacting flow. The key parameters used in the simulation are presented in Table S2. The current density expressions for the anode and cathode in the fuel cell interface are based on the Butler–Volmer equation, as depicted below:

Hydrogen oxidation reaction:

$$i_a = i_{0,ref,a} \left( \frac{p_{H_2}}{p_{ref}} \exp\left( \frac{\alpha_{a.a}}{RT} F\eta_a \right) - \exp\left( \frac{\alpha_{c.a}}{RT} F\eta_a \right) \right) \quad (2)$$

$i_{0,ref,a}$ represents the reference current density at the anode. $p_{H_2}$ denotes the partial pressure of hydrogen gas, $\alpha_a$ and $\alpha_c$ are the charge transfer coefficients. R stands for the gas constant, T is the temperature, and F is the Faraday constant. $p_{ref}$ is the reference pressure set at 1 atm. $\eta_a$ is the overpotential relative to the equilibrium potential of the reaction at atmospheric pressures of the reacting gases.

Oxygen reduction reaction:

$$i_c = i_{0,ref,c} \left( \left( \frac{p_{H_2O}}{p_{ref}} \right)^2 \exp\left( \frac{\alpha_{a.c}}{RT} F\eta_c \right) - \frac{p_{O_2}}{p_{ref}} \exp\left( -\frac{\alpha_{c.c}}{RT} F\eta_c \right) \right) \quad (3)$$

$p_{O_2}$ and $p_{H_2O}$ respectively denote the local partial pressures of oxygen and water. The charge transfer coefficients adhere to the law of mass action:

$$\alpha_a + \alpha_c = n \quad (4)$$

n represents the number of electrons transferred. The electrochemical reaction rate can be expressed by correlating the rate of the reaction with the activation overpotential:

$$\eta = \phi_s - \phi_l - E_{eq} \quad (5)$$

The expression $\phi_s - \phi_l$ denotes the disparity between the potentials of the electrode and electrolyte phases. $E_{eq}$ is the equilibrium potential, directly associated with the variation in Gibbs free energy of the reacting species.

The Fuel Cell module in COMSOL not only facilitates the computation of the aforementioned electrochemical reactions but also enables the calculation of mass transfer based on the Maxwell-Stefan equation in hydrogen fuel cell interface[64]. Momentum transfer studied by Free and Porous Media Flow interfaces, on the other hand,

necessitates the utilization of compressible Navier-Stokes equations in the pore phase, and the Brinkman equations and Darcy's Law in the mixture phase. The relevant equations and explanations are presented in the supporting information.

The specific boundaries vary among different MEAs due to the complex three-dimensional structure. The electronic potential at the GDL boundaries facing the flow pattern ribs is set to zero, while the corresponding boundaries on the cathode side are set to the cell potential. All remaining external boundaries are electrically isolated. No ship conditions are applied for all wall boundaries.

## Data availability

Source data are provided with this paper. The datasets generated during the current study are available in the UCL Research Data Repository, https://doi.org/10.5522/04/27303723.v1 (CC BY-NC-SA 4.0) Source data are provided with this paper.

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

## Acknowledgements

T.S.M., S.M.H., and P.R.S acknowledge support from the UK Engineering and Physical Sciences Research Council (EPSRC) - EP/W03395X/1, EP/X023656/1, EP/W033321/1, EP/P009050/1. Electron Microscopy access was supported by the Henry Royce Institute for Advanced Materials, funded through EPSRC grants EP/R00661X/1, EP/S019367/1, EP/P025021/1 and EP/P025498/1. S.J.H. thanks the European Research Council (ERC) under the European Union's Horizon 2020 research and innovation programme (Grant ERC-2016-STG-EvoluTEM- 715502). X. Lu acknowledges funding from Engineering and Physical Sciences Research Council (EP/X000702/1).

## Author contributions

J.C., S.M.H., and T.S.M. conceived the study; L.W. and H.G. processed the membrane with laser; J.C., H.Z., and Y.L. did the material characterization; J.C. did the fuel cell test; J.C. and W.D. did the CT scan; M.R. and S.J.H. did the FIB-SEM; P.R.S. advised on the data analysis and result interpretation; J.C. and X.L. did the simulation; J.C. drafted the manuscript and all co-authors reviewed the manuscript.

## Competing interests

The authors declare no competing interests.
