## [Peer Review File · Nature Communications]

REVIEWER COMMENTS

Reviewer #1 (Remarks to the Author):

This paper explored the structure and mass transfer in MEAs with PA doping and Laser-induced modification of membranes using the image-based three-dimensional multi-physics method. Several approaches including SEM, EDX, CT, and CFD simulation were employed simultaneously to observe the microscale structures and mass transfer. This paper is interesting and meaningful. However, several major concerns should be addressed carefully.

1) This paper studied the HT-PEMFC with PBI membrane. In the Introduction section, the authors should discuss the recent research progress on HT-PEMFC, including the performance, durability, material development, etc, to provide more useful information for readers.

2) The authors used the micro-CT to observe the MEA structure. Generally, the CL has many nanoscale pores that can not be detected by micro-CT, which also play a significant role in the mass transfer in CLs. However, I didn't see how the authors deal with this serious issue.

3) As for the CFD simulation via COMSOL, another major concern is existed. Again, the MEA, especially the CL, has nanoscale pores, where the Knudsen number is larger, and the continuity hypothesis may be invalid. In this case, the macroscale CFD simulation model is not inapplicable.

4) Moreover, the errors between simulation and experiment results as well as the major source of errors should be provided. Also, the model assumptions should be provided.

Reviewer #2 (Remarks to the Author):

In this manuscript entitled "Optimising High-Temperature Proton Exchange Membrane Fuel Cell Three-Phase Interfaces: Scalable Membrane Modification via Picosecond Laser Scribing", Chen et al. report their studies on the optimization of three-phase interfaces in PBI-based MEA using laser processing and phosphoric acid (PA) modification. The authors used X-ray μ -CT and image-based modelling to understand the difference between MEAs based on different membranes after treatments. The following concerns should be addressed before further consideration for publication.

(1) What are the key findings of this work? The key finding should be stated more clearly and concisely in the abstract and conclusion.

(2) PA modification is a well-known strategy to improve the performance of PBI membranes. The electrochemical performance of MEA based on PBI without laser processing but after PA modification is particularly important for comparison, which is missing in this work.

(3) Further characterization of the properties of different membranes, such as ionic conductivity, mechanical behavior, etc. should be carried out.

(4) The electrochemical performance of MEA based on PBI should be given in Fig. S1 for comparison.

(5) What is the depth of the regions after laser processing as shown in Fig. 1b(ii)?

(6) Page 8 "The impact of laser etching on the dry membrane is evident in its effect on the membrane's

permeability to H₂. However, the doping of PA effectively mitigates this influence. ” Is there any experiment evidence to support this statement? How can PA contribute to increase in OCV?

(7)Page 6 “whereas the laser-treated PBI membrane exhibits noticeable protrusions due to graphene foam formation at the laser tracks, likely caused by the detachment of graphene layers.” Is there any experiment evidence to support the formation of graphene?

(8)In Fig. 2b,c, the electrochemical performance of MEA LPAc and LPAa was improved after AST. What is the reason behind this phenomenon? Will the performance be further improved after prolonged test? A long-term stability test should be given for at least 100 h.

(9)The Nyquist plots (Fig.2d,e) should be isotropic, which means that the scales for X axis and Y axis should be fully the same (i.e., the length of 1 ohm cm² in X axis is fully identical to the length of 1 ohm cm² in Y axis)

(10)In conclusion, the authors stated “...a membrane interface capable of controlling PA leaching was created.” Is there any experimental evidence to support this statement?

Reviewer #3 (Remarks to the Author):

In this work, the authors systematically investigated the impact of membrane modified by picosecond laser scribing on the performance of HT-PEMFCs, and more importantly, they developed an image-based 3D multi-physics and multiphase HT-PEMFC MEA based on X-ray micro computed tomography (μ -CT) data, to reveal the underlying understandings. Overall, the manuscript is interesting and well organized. I would recommend it for publication after addressing the following issues.

1. I am wondering if the authors could control the thickness of graphene/graphene oxide by laser scribing? And how does it affect the performance of HT-PEMFCs?
2. The authors used air in the simulation, which is different from the oxygen used in the experiment, in order to more clearly illustrate the impact of material migration on gas distribution. I am wondering, why air can illustrate the impact of material migration more clearly?
3. The authors stated that “laser treated membranes (particularly those with graphene on the cathode side) gradually enhances the electrochemical reaction performance within the electrode, consequently improving the performance and durability of the MEA, despite the laser induction being on the membrane surface”, however, the performance of LPAac is still inferior to that of PBI after AST (Figure 2). This indicates that the membrane treated by laser on both sides results in poorer fuel cell performance, hence I suggest the authors to revise the above statement. Moreover, if the authors extend the AST, would the performance of LPAac further increase and eventually surpass the PBI membrane?
4. How did authors calculate the Ohmic component, charge transfer and mass transfer resistances in Fig.2 and Fig.S1? The authors should provide the equivalent circuit models. Moreover, in Fig. S1c, polarization curves after AST, why does the power density and voltage of La suddenly decrease? Is there an explanation for this?
5. The membrane curvature of LPAac is similar to that of PBI, both of which are smaller than LPAa and LPAc. The authors explained that this is because of bending is driven by a difference in the tension between the two sides of the membrane. However, for the membrane treated by laser on one side, i.e., LPAa and LPAc, the curvature of LPAa is much smaller than LPAc, can the authors explain for this?
6. LPAac exhibits the best oxygen and hydrogen flux in the CL layer, but its performance is inferior to both LPAa and LPAc. Is there an explanation for this?

7. There are issues with the axis labels in Fig. 4b, the vertical axis is missing. Also, there are errors in the figures, some of the labels (μm) are obscured in the Figures 5-8 and Supplementary Figures. The authors should carefully revise them.

8. In the Discussion section, there are many uncertain claims made in the paper, such as “may” etc., I would suggest the authors provide a clearer explanation or evidence.

We would first like to thank the reviewers and editors for their input, we greatly appreciate their considered feedback. We have revised the manuscript in line with the comments provided. Our responses to the questions raised can be found below.

Additions to the manuscript have been highlighted (see highlighted MS).

Reviewer #1:

1) This paper studied the HT-PEMFC with PBI membrane. In the Introduction section, the authors should discuss the recent research progress on HT-PEMFC, including the performance, durability, material development, etc, to provide more useful information for readers.

Response:

We have now updated the introduction with recent developments related to HT-PEMFCs, particularly focusing on membrane advancements. We have worked to keep this overview concise to ensure the introduction did not become overly long.

MS changes:

One area of particular research focus for HT-PEMFCs is the development of membrane systems. Various polymers have been developed to enhance membrane proton conductivity of HT-PEMFCs, such as polyethersulfone–polyvinylpyrrolidone (PES–PVP),¹⁴ while others, like polymers of intrinsic microporosity (PIMs)¹⁵, have been employed to improve the longevity of HT-PEMFCs under complex conditions. However, advancements in the most developed polymer for HT-PEMFCs, polybenzimidazole (PBI), remains a focal point in this field. Modifications of PBI, including functionalization^{16–18}, crosslinking, grafting¹⁹, and doping²⁰, have been extensively explored. HT-PEMFCs utilizing different membranes generally achieve peak power densities in the range of 400-700 mW cm⁻²^{14,21}. However, these membranes typically exhibit varying degrees of performance degradation during durability tests^{17,19,20}, with only a few studies demonstrating stable performance²².

2) The authors used the micro-CT to observe the MEA structure. Generally, the CL has many nanoscale pores that can not be detected by micro-CT, which also play a significant role in the mass transfer in CLs. However, I didn't see how the authors deal with this serious issue.

Response: Please see below (comment 3)

3) As for the CFD simulation via COMSOL, another major concern is existed. Again, the MEA, especially the CL, has nanoscale pores, where the Knudsen number is larger, and the continuity hypothesis may be invalid. In this case, the macroscale CFD simulation model is not inapplicable.

Response:

As comments 2 and 3 are interrelated we have presented a combined response in which we show new and additional experimental data.

As pointed out by the reviewer, the micron-scale model we employed may not fully capture the nuances of nanoscale features, particularly the pore structure within the CL. We did attempt to utilize the Zeiss Xradia 810 Ultra for nanoscale scanning, however nano-CT scanning is inherently limited to a specific region and cannot cover all components of the MEA. Even if full-field scanning were achievable, the data volume would likely exceed the mesh construction and simulation capabilities of conventional supercomputers, limiting its applicability. Conversely, micro-CT scanning can encompass all components of the MEA and remains within computational capacity limits, yet still offering significant advances previously published works utilizing 2D or block-based simplified 3D models.

We previously investigated the convergence of the model by coupling the fuel cell module with the free and porous media flow module. This was achieved through a composite approach that utilized the actual three-dimensional structure of the GDL and the parameter settings in the CL. The pore-wall friction forces utilize a wall diffusivity, which is set to Knudsen diffusivity by default. The Knudsen diffusivity is derived from kinetic gas theory by relating the mean free path of gas molecules to the pore diameter. The gas diffusivity defined in the Gas Phase node is multiplied by a correction factor to account for the gas pore volume fraction being less than 1. This correction can be applied using either the Bruggeman correlation, Tortuosity, or a user-defined correlation. For Tortuosity, we can select Isotropic to define a

scalar value, or Diagonal or Symmetric to define anisotropic tensor values. The implementation of this composite model relies on several assumptions and default parameters. For instance, it was assumed that the gas permeability of the CL is one-fifth that of the GDL. This will be elaborated in detail in the response to comments 4. However, we recognise that this assumption does deviate from the actual conditions to some extent. Therefore, we have since conducted supplementary experiments to investigate the nanopores within the CL by employing Focused Ion Beam Scanning Electron Microscopy (FIB-SEM) to achieve a voxel resolution of in the nanometre range. This high-resolution scanning allowed us to investigate the nanopores within the catalyst layer, which subsequently enabled us to conduct permeability simulations and provide parameters for our micron-scale model by utilizing simulations in Avizo, providing more accurate parameters for COMSOL. The FIB-SEM scanning results and permeability simulations are illustrated in the figure below and also presented in the manuscript's Supporting Information Fig. S15 a and b.

Fig. S 15. 3D structure of CL. a FIB-SEM images of CL. **b** Illuminated streamlines of absolute permeability in the CL.

Based on this FIB-SEM scanning data we have adjusted the permeability ratio between the GDL (Gas Diffusion Layer) and the catalyst layer from the original 5:1 to 6.25:1.

Importantly, from the updated simulation results, it is evident that this change has a minimal impact on the polarization curve from OCV (Open Circuit Voltage) to 0.4 V. However, it does affect the gas flow and voltage distribution. We have therefore updated the manuscript to reflect these new simulation results.

Fig. 6. Images depicting the simulation results of LPAa varying with voltage. **a** Total flux streamline of O_2 . **b** Electrolyte potential (arrow volume: electrolyte current density vector), **c** electrode potential with respect to ground (arrow volume: electrode current density vector).

Fig. 7. Visualization simulation and statistics of gas distribution and velocity at 0.4 V. a Total flux mole fraction surface and streamline of H₂. **b** Total flux mole fraction surface and streamline of O₂. **c** Statistical distribution of the molar fractions of oxygen and hydrogen in the catalyst layer. The green area and purple bars represent the distribution range of the molar fraction of oxygen and hydrogen, respectively. The black and red bars correspond to the error bars indicating the minimum and maximum values of the molar fraction of oxygen, respectively. **d** velocity magnitude slices of O₂.

Fig. 8. Visualization simulation of potential, current density and water distribution at 0.4V. a Electrolyte potential (arrow volume: electrolyte current density vector). **b** electrode potential with respect to ground (arrow volume: electrode current density vector). **c** water mole fraction distribution surface

4) Moreover, the errors between simulation and experiment results as well as the major source of errors should be provided. Also, the model assumptions should be provided.

Response:

While we have made every effort to replicate real-world conditions through our model, limitations in experimental and simulation precision, assumptions, and computational capabilities certainly introduce discrepancies between the model and actual conditions which we agree we must acknowledge. These factors have been further explained in the manuscript as follows:

MS changes:

Based on the parameter settings reported in previous literature for simplified 2D or 3D models of HT-PEMFCs^{33–35,64,65}, it is assumed that the permeability ratio between the nanopores in the CL or mixture phase and the micron-scale pores in the GDL remains consistent. Focused Ion Beam Scanning Electron Microscopy (FIB-SEM) was employed to investigate the three-dimensional spatial distribution of nanopores within the CL and to conduct permeability simulations. The FIB-SEM scanning results and permeability simulation diagrams are shown in Fig. S12 a and b. By calculating the average permeability in the CL and GDL, this ratio was determined to be 6.25. However, due to the impracticality of extending nanoscale scanning to all components of the entire MEA, the CL was modelled as a porous homogenous layer. This simplification is one of the primary sources of error in this simulation approach.

In addition to the aforementioned factors, discrepancies between simulation and actual conditions can also arise from the precision of the scanning process, the accuracy of segmentation, and the fidelity of finite element meshing.

Reviewer #2:

(1) What are the key findings of this work? The key finding should be stated more clearly and concisely in the abstract and conclusion.

Response:

We have added the following content to the abstract and conclusion to clarify the key findings of this work.

MS changes:

Abstract

Experimental and simulation analyses demonstrate that MEAs incorporating a PA-doped membrane induced by laser treatment on the cathode side achieves the highest peak power density (817.2 mW cm⁻²) after AST, due to its optimal performance at the three-phase interface.

Conclusion

The localized foam-like graphene oxide structures formed by laser treatment can affect the penetration of PA and result in a sawtooth distribution of PA within the catalyst layer, thereby increasing the triple-phase boundary. Additionally, the asymmetry between the two sides of the membrane caused by one-sided laser treatment can lead to increased membrane curvature, further enhancing the triple-phase boundary. Moreover, the catalyst migration resulting from this effect, along with the differences in the pore network of the GDL, further contributes to the variations in the triple-phase boundary.

(2) PA modification is a well-known strategy to improve the performance of PBI membranes. The electrochemical performance of MEA based on PBI without laser processing but after PA modification is particularly important for comparison, which is missing in this work.

Response:

As highlighted by the reviewer, the modification of PA plays a crucial role in the field of HT-PEMFCs. The term "PA modification" is relatively broad, encompassing various factors such as

adjustments to the polymer's molecular weight, the temperature and duration of PA doping, and the resulting acid doping level. It is believed that the reviewer is specifically referring to the treatment methods applied to the membrane during PA doping.

To eliminate any potential variability in membrane properties that could affect the impact of laser treatment, we utilized commercially available and performance-stable PBI membranes. During PA doping, we strictly adhered to the handling and doping protocols provided by Fumatech. By following these established protocols, we ensured that the PBI membrane exhibited optimal performance. Consequently, no further modifications were made to the membrane's doping process.

This approach was taken to maintain consistency and reliability in our results, ensuring that the observed effects were attributable to the laser treatment rather than variations in the membrane's doping process.

(3) Further characterization of the properties of different membranes, such as ionic conductivity, mechanical behavior, etc. should be carried out.

Response:

We have supplemented the supporting information with details on the conductivity and strain-stress testing, and have analyzed these aspects in the main text. The relevant figures and analyses are presented as follows.

MS changes:

Fig. S5. Proton conductivity of PA doped PBI membrane and laser-treated PBI membranes.

The average proton conductivity of different membranes is shown in Fig. S5. Regardless of whether laser treatment was applied before or after PA doping, the proton conductivity of the membranes did not improve compared to untreated PBI, generally a lower conductivity was observed. Membranes that underwent double-sided laser treatment exhibited lower conductivity compared to those treated on a single side. Importantly however, membranes subjected to laser treatment after PA doping demonstrated higher proton conductivity than those treated with laser before PA doping.

Fig. S12. Stress–strain curves of PBI and laser induced PA doped PBI membrane.

Stress-strain curves for pure PBI and PA-doped PBI membranes subjected to single-sided and double-sided laser induction are shown in Fig. S12. The linear regions of the different curves coincide, indicating that the Young's modulus remains relatively unchanged. However, laser induction significantly reduces the yield strength. Notably, the tensile strength of membranes subjected to single-sided laser induction decreases, making them more prone to deformation. This phenomenon is attributed to the differences between the two sides of the membrane and the uneven stress distribution caused by the roughness of the single-treated surface. In contrast, membranes subjected to double-sided laser induction exhibit an increase in tensile strength. Additionally, both single-sided and double-sided laser-induced membranes show an improvement in elongation at break. The impact of laser induction on the mechanical properties of PA-doped PBI membranes could result in varying degrees of deformation within the MEA.

(4) The electrochemical performance of MEA based on PBI should be given in Fig. S1 for comparison.

Response:

We have added the performance data related to PBI in what was Fig S1 but is now S2, for comparison. The revised figure is shown below.

Fig. S 2. Electrochemical characterization of MEAs based on different membranes. a AST. b polarization curves before AST. c polarization curves after AST, d EIS curves before AST, e EIS curves after AST

(5) What is the depth of the regions after laser processing as shown in Fig. 1b(ii)?

Response:

We have included in the Supplementary Information an image with depth profile data, as shown in Fig. S1.

Fig. S 1. Optical microscopy of laser-induced PBI membrane with depth profile.

(6) Page 8 “The impact of laser etching on the dry membrane is evident in its effect on the membrane’s permeability to H₂. However, the doping of PA effectively mitigates this influence. ” Is there any experiment evidence to support this statement? How can PA contribute to increase in OCV?

Response:

We apologise for any ambiguity in our previous draft of this manuscript. We did not mean that PA affects OCV directly; rather, the membrane doped with PA prior to laser induction demonstrates a higher OCV in its MEA compared to the membrane that undergoes laser induction before PA doping. This suggests that PA doping reduces the impact of hydrogen crossover caused by the laser treatment. We have revised the relevant statements in the original text and supplemented with LSV data to further demonstrate the differences in hydrogen crossover as below:

MS changes:

Therefore, it can be seen that performing PA doping prior to laser induction is more effective in reducing the impact of laser induction on gas crossover compared to doping after laser induction. This conclusion is further validated by the linear sweep voltammetry (LSV) curves

of Lac and LPAac in Fig. S 3. The calculated hydrogen crossover values for Lac and LPAac according to LSV are 0.71×10^{-8} and 1.53×10^{-8} , respectively.

Fig. S 3. LSV curves of LPAac and Lac.

(7) Page 6 “whereas the laser-treated PBI membrane exhibits noticeable protrusions due to graphene foam formation at the laser tracks, likely caused by the detachment of graphene layers.” Is there any experiment evidence to support the formation of graphene?

Response:

We apologise, the wording that we used in the original text was slightly inaccurate; it should specifically refer to graphene oxide rather than graphene. We have already corrected this in the text. The production of graphene or graphene oxide through laser treatment of polymers has been reported in the literature (DOI: 10.1038/ncomms6714). However, the goal of these studies was to continuously induce the polymer with a laser to convert the entire polymer into graphene material. The distinction of this work lies in using laser treatment on the polymer surface to generate thin graphene oxide layer, ensuring that hydrogen gas cross-linking does not occur. Unlike large-scale production methods that use polymers to manufacture graphene, our surface laser-induced process produces only a thin layer of graphene oxide, making it difficult to separate from the polymer. Nonetheless, we have attempted to confirm that the substance generated on the surface is indeed graphene oxide.

Firstly, as shown in Fig. 1c and d, the EDX analysis reveals the presence of oxygen on the membrane surface after laser treatment, which is not inherent to PBI, indicating surface oxidation. Since the generated graphene oxide is challenging to separate from the membrane, we used the laser-treated membrane surface for Raman testing. Although the presence of the membrane creates significant background and fluorescence effects in the characterization of graphene oxide, the characteristic D and G peaks of graphene oxide are clearly visible in the Raman spectra compared to that of PBI. We have identified the positions of the D and G peaks on the Raman spectra. The G peak appears as a broad peak characteristic of graphene oxide, rather than the narrow peak typical of graphite. We believe these findings, combined with the aforementioned results, confirm the formation of graphene oxide.

Fig. 1. Laser-induced membrane. c, EDX images, (i) SEM image for EDX, (ii) C element, (iii) N element and (iv) O element, **d**, EDX spectrum and elements content, **e**, Raman spectra of PBI membrane and Laser-induced PBI membrane

(8) In Fig. 2b,c, the electrochemical performance of MEA LPAc and LPAa was improved after AST. What is the reason behind this phenomenon? Will the performance be further improved after prolonged test? A long-term stability test should be given for at least 100 h.

Response:

The primary difference between HT-PEMFCs and conventional low-temperature fuel cells is that HT-PEMFCs use phosphoric acid (PA) as the proton-conducting medium. HT-PEMFCs typically use PTFE, which is non-proton-conductive, as the catalyst binder instead of ionomers. Consequently, the establishment of the triple-phase boundary in the catalyst depends on the leaching of PA from the membrane into the catalyst layer. The extent of the triple-phase boundary area relies on the distribution of PA (electrolyte) within the catalyst layer. An optimal amount of PA leaching creates a well-formed triple-phase boundary, whereas excessive PA leaching can lead to complete coverage of the catalyst layer, resulting in performance degradation.

Laser induction slows the PA leaching (which is further demonstrated with supplementary experiments, discussed in response to comment 10 below). Therefore, during the Accelerated Stress Test (AST), the triple-phase boundary undergoes a gradual optimization process due to the slow leaching of PA. In contrast, the Membrane Electrode Assemblies (MEAs) based on pure PBI membranes experience rapid PA leaching, causing the catalyst to become flooded. Of course, this difference is not only due to the impact of laser-induced graphene oxide on PA leaching but also the asymmetric laser treatment, which causes variations on both sides of the membrane and results in changes in membrane curvature, thereby affecting PA distribution. Additionally, considering the different mechanisms of PA leaching at the cathode and anode, where the anode is influenced by the electrochemical pump effect and the cathode is mainly affected by water production, these combined factors contribute to the performance improvement of LPAc over LPAa after AST.

As noted by the reviewer, durability tests for fuel cells typically extend beyond 100 hours, but these are mostly conducted under constant current conditions. In this study, the Accelerated Stress Test (AST) method is designed to expedite the leaching of PA, resulting in a degradation rate that is five times faster than that observed in constant current tests (DOI: 10.1002/fuce.201500160). Therefore, the AST used in this work is equivalent to over 150

hours of constant current testing. To illustrate the differences between various MEAs during the AST process, stacked AST plots from the first cycle to the fifth cycle, with each cycle spanning six hours, are shown in Fig. S4. As seen in Fig. S4, the performance of LPAa in the fifth cycle began to decline compared to the fourth cycle, whereas LPAc remained relatively stable. This further demonstrates the superiority of LPAc over LPAa.

Fig. S4. Stacked AST plots from the first cycle to the fifth cycle (repeating chronopotentiometry between 0.6 A cm⁻² for 4 min and 1 A cm⁻² for 16 min, running OCV for 10 min every 6 hr)

(9) The Nyquist plots (Fig.2d,e) should be isotropic, which means that the scales for X axis and Y axis should be fully the same (i.e., the length of 1 ohm cm² in X axis is fully identical to the length of 1 ohm cm² in Y axis)

Response:

We apologise for this error. We have modified the x and y axes in Figures 2d and 2e to maintain isotropy, as shown in the revised figures below.

MS changes:

(10) In conclusion, the authors stated “...a membrane interface capable of controlling PA leaching was created.” Is there any experimental evidence to support this statement?

Response:

We have added the acid retention test to support this statement. The acid retention test (DOI:10.1016/J.POLYMER.2015.03.040) involved suspending the PA-doped membranes above boiling water for 5 hours. The weight of the membranes was recorded every hour to measure the amount of acid leached. The time dependence of weight loss ratio of acid in different membranes is shown in Fig.S10.

MS changes:

This is further illustrated in Fig. S10, which shows the time dependence of the weight loss ratio of acid.

Fig. S 10. Time dependence of weight loss ratio of acid in PBI, one side laser treated PBI and two side laser treated PBI.

Reviewer #3:

1. I am wondering if the authors could control the thickness of graphene/graphene oxide by laser scribing? And how does it affect the performance of HT-PEMFCs?

Response:

We agree with the reviewer that the degree of laser induction would be of great interest. However, we found it challenging to use the thickness of the laser scribing as a variable. First, the doping of phosphoric acid makes it difficult to precisely detect the depth induced by laser induction. Second, it is challenging to accurately correlate the laser parameters with the linearly incremental laser-induced depths. Therefore, we investigated the effects of different laser fluences, which represent the varying impacts of laser energy on the membrane. We have included comparisons of LPAC (which shows the best performance) treated with laser fluences of 15.8 mJ cm^{-2} , 21.2 mJ cm^{-2} , and 28.3 mJ cm^{-2} , as shown below, for further comparison.

MS changes:

The laser-induced membrane discussed above was prepared with a laser fluence of 21.2 mJ cm^{-2} . To elucidate the differences in electrochemical performance under varying laser energies, tests were conducted on LPAC membranes treated with laser fluences of 15.8 mJ cm^{-2} and 28.3 mJ cm^{-2} . The membrane surface treated by different laser energies is shown in Fig. S6a. The polarization curves and power density curves of PBI and LPAC with different laser fluences are shown in Fig. S6b. As shown in Fig. S6b, the impact of lower laser energy on the PBI is minimal, whereas higher laser energy can lead to hydrogen crossover, which is reflected in the lower OCV. This, in turn, affects the performance improvement. Therefore, precise control of laser energy is crucial for optimizing performance enhancement.

Fig. S 6. LPaC with different laser fluences of 15.8 mJ cm⁻², 21.2 mJ cm⁻², and 28.3 mJ cm⁻², a Membrane surface. **b** Polarization curves and power density curves of LPaC 160°C, anode: H₂ ($\lambda=1.2$), cathode: O₂ ($\lambda=2.0$).

2. The authors used air in the simulation, which is different from the oxygen used in the experiment, in order to more clearly illustrate the impact of material migration on gas distribution. I am wondering, why air can illustrate the impact of material migration more clearly?

Response:

In our work we employed a multiphase, multiphysics coupling approach that integrates computational fluid dynamics (CFD) with electrochemical reactions. This approach accounts

for the differences among various models, which arise from membrane deformation, catalyst migration, and phosphoric acid (PA) transport, leading to variations in 3D structures, pore networks, and permeability. Simulating with air can reduce the mole fraction of oxygen at the cathode, thereby further illustrating the impact of these factors on mass transfer from an electrochemical performance perspective. This reduction in oxygen content under caustic conditions results in a decline in performance, highlighting the aforementioned influences.

MS changes:

The simulation used air instead of pure oxygen to more clearly illustrate the impact of material migration on gas distribution. By reducing the mole fraction of oxygen at the cathode, the effects of membrane deformation, catalyst migration, and PA transport on mass transfer are highlighted, showing a performance decline under caustic conditions, different to the oxygen used in the experiments.

3. The authors stated that “laser treated membranes (particularly those with graphene on the cathode side) gradually enhances the electrochemical reaction performance within the electrode, consequently improving the performance and durability of the MEA, despite the laser induction being on the membrane surface”, however, the performance of LPAac is still inferior to that of PBI after AST (Figure 2). This indicates that the membrane treated by laser on both sides results in poorer fuel cell performance, hence I suggest the authors to revise the above statement. Moreover, if the authors extend the AST, would the performance of LPAac further increase and eventually surpass the PBI membrane?

Response:

We have now revised the statement

MS changes:

one side laser treated membranes (particularly those with graphene on the cathode side) gradually enhances the electrochemical reaction performance within the electrode

To better illustrate the voltage variations during the AST process, we have opted to stack different cycles in six-hour intervals, as shown in Figure S4. It is evident that LPAac exhibits a significant increase during the first cycle and subsequently remains more stable compared to other MEAs. However, by the fifth cycle, LPAac begins to show a decline trend. Consequently, the performance of LPAac is unlikely to improve further.

Fig. S4. Stacked AST plots from the first cycle to the fifth cycle (repeating chronopotentiometry between 0.6 A cm⁻² for 4 min and 1 A cm⁻² for 16 min, running OCV for 10 min every 6 hr)

4. How did authors calculate the Ohmic component, charge transfer and mass transfer resistances in Fig.2 and Fig.S1? The authors should provide the equivalent circuit models. Moreover, in Fig. S1c, polarization curves after AST, why does the power density and voltage of La suddenly decrease? Is there an explanation for this?

Response:

The equivalent circuit used for fitting and the definition of parameters are shown in Table S1.

Table S1. The equivalent circuit and the definition of parameters

Equivalent circuit	L_{wires} (H)	Inductance of cables
R_{Ω} (ohm)	Ohmic resistance
R_c (ohm)	Charge transfer resistance
R_m (ohm)	Mass transfer resistance
Y_o (S s ^a)	Constant phase element
a	Dimensionless exponent

After AST, the polarisation curve of La may certainly does not exhibit a perfect polarisation curve, yet we deliberately retained this curve to illustrate observed phenomena during experiments. In the original text, we mentioned that laser induction on the anode side leads to accelerated PA leaching on the cathode side, a phenomenon already substantiated in the manuscript. The sudden drop in La's polarisation curve with increasing current density is attributed to rapid cathode PA leaching, causing water enrichment and subsequent catalyst flooding. This closely resembles water flooding in low-temperature fuel cells. We have provided additional explanatory details in the manuscript as follows:

MS changes:

The polarization curve of La shown in Fig. S2 exhibits a sudden drop in polarization after AST at intermediate current densities, also reflecting the reaction induced by laser treatment on the anode side. The instantaneous accumulation of a large amount of PA and water on the cathode side leads to catalyst flooding, thereby instantaneously obstructing mass transfer.

5. The membrane curvature of LPAac is similar to that of PBI, both of which are smaller than LPAa and LPAc. The authors explained that this is because of bending is driven by a difference in the tension between the two sides of the membrane. However, for the membrane treated by laser on one side, i.e., LPAa and LPAc, the curvature of LPAa is much smaller than LPAc, can the authors explain for this?

Response:

From the stress-strain curves now found in Fig. S12, it is evident that unilateral laser treatment indeed reduces the tensile stress of the membrane, making it more prone to deformation. A more detailed analysis is presented in response to comment 8.

We have added the following content to the manuscript to explain the differences in membrane curvature between LPAa and LPAc

MS changes:

Although both LPAa and LPAc membranes were subjected to unilateral laser treatment, several differences exist between the cathode and anode in HT-PEMFCs. The cathode involves water generation, whereas the anode does not. Additionally, the mechanisms of PA leaching differ between the anode and cathode. Different laser treatments can lead to varying degrees of PA leaching, and the PA content affects mechanical properties. The combined thermal effects and compositional migration on both sides contribute to the varied curvature observed in LPAa and LPAc membranes.

Fig. S 12. Stress–strain curves of PBI and laser induced PA doped PBI membrane.

6. LPAac exhibits the best oxygen and hydrogen flux in the CL layer, but its performance is inferior to both LPAa and LPAc. Is there an explanation for this?

Response:

LPAac has a smooth membrane and a pristine MEA electrode structure which seems ideal. However, this resembles the state of a non-activated MEA. HT-PEMFCs have unique characteristics: their catalyst layers do not use ionomers but PTFE as a binder. Proton conduction in the catalyst requires a certain amount of PA infiltration. The excellent oxygen and hydrogen flux in LPAac indicates that the CL contains very little PA. Even though hydrogen or oxygen may react on the catalyst surface, the reaction cannot propagate efficiently. This explains why, despite good material transport, the $R_{\Delta(c+m)}$ in LPAac remain high. Additionally, laser induction has been shown to slightly reduce proton conductivity, and the impact is greater when both sides are treated than when only one side is treated.

This work aims to demonstrate the superiority of seemingly "imperfect" structures. For HT-PEMFCs, we argue that good performance does not stem from a seemingly "perfect" three-dimensional structure or the optimization of a single phase, but from the coupling and synergy of multiple phases and physical fields.

7. There are issues with the axis labels in Fig. 4b, the vertical axis is missing. Also, there are errors in the figures, some of the labels (μm) are obscured in the Figures 5-8 and Supplementary Figures. The authors should carefully revise them.

Response:

We have corrected the errors in the relevant figures as follows:

Fig. 4. Pore structure and spatial distribution in MEAs after AST. b The statistical distribution plot of the average curvature of the membrane.

Fig. 6. Images depicting the simulation results of LPAa varying with voltage. **a** Total flux streamline of O_2 . **b** Electrolyte potential (arrow volume: electrolyte current density vector), **c** electrode potential with respect to ground (arrow volume: electrode current density vector).

Fig. 7. Visualization simulation and statistics of gas distribution and velocity at 0.4 V. a Total flux mole fraction surface and streamline of H₂. **b** Total flux mole fraction surface and streamline of O₂. **c** Statistical distribution of the molar fractions of oxygen and hydrogen in the catalyst layer. The green area and purple bars represent the distribution range of the molar fraction of oxygen and hydrogen, respectively. The black and red bars correspond to the error bars indicating the minimum and maximum values of the molar fraction of oxygen, respectively. **d** velocity magnitude slices of O₂.

Fig. 8. Visualization simulation of potential, current density and water distribution at 0.4V.
a Electrolyte potential (arrow volume: electrolyte current density vector). **b** electrode potential with respect to ground (arrow volume: electrode current density vector). **c** water mole fraction distribution surface

8. In the Discussion section, there are many uncertain claims made in the paper, such as “may” etc., I would suggest the authors provide a clearer explanation or evidence.

Response:

We have now supplemented the study with FIB-SEM to demonstrate the presence of nanopores, stress-strain tests to assess the impact of laser induction on mechanical properties, LSV to verify hydrogen crossover in different membranes, and PA retention tests to illustrate the differences in PA leaching among various membranes. The relevant results are summarised below:

MS changes:

By employing Focused Ion Beam Scanning Electron Microscopy (FIB-SEM) to achieve a voxel resolution of nanometres. This high-resolution scanning allowed us to investigate the nanopores within the catalyst layer, which subsequently enabled us to conduct permeability simulations and provide parameters for our micron-scale model. The FIB-SEM scanning results and permeability simulations are illustrated in the figure below and also presented in the manuscript's Supporting Information Fig. S15 a and b.

Fig. S15. 3D structure of CL. a FIB-SEM images of CL. **b** Illuminated streamlines of absolute permeability in the CL.

Fig. S 12. Stress–strain curves of PBI and laser induced PA doped PBI membrane.

Stress-strain curves for pure PBI and PA-doped PBI membranes subjected to single-sided and double-sided laser induction are shown in Fig. S12. The linear regions of the different curves coincide, indicating that the Young's modulus remains relatively unchanged. However, laser induction significantly reduces the yield strength. Notably, the tensile strength of membranes subjected to single-sided laser induction decreases, making them more prone to deformation. This phenomenon is attributed to the differences between the two sides of the membrane and the uneven stress distribution caused by the roughness of the single-treated surface. In contrast, membranes subjected to double-sided laser induction exhibit an increase in tensile strength. Additionally, both single-sided and double-sided laser-induced membranes show an improvement in elongation at break. The impact of laser induction on the mechanical properties of PA-doped PBI membranes could result in varying degrees of deformation within the MEA.

It can be seen that performing PA doping prior to laser induction is more effective in reducing the impact of laser induction on gas crossover compared to doping after laser induction. This conclusion is further validated by the linear sweep voltammetry (LSV) curves of Lac and LPAac in Fig. S 3. The calculated hydrogen crossover values for Lac and LPAac according to LSV are 0.71×10^{-8} and 1.53×10^{-8} , respectively.

The LSV measurements were conducted using a GAMRY Reference 3000. During the LSV measurement, the voltage was increased from 0 to 550 mV at a scan rate of 1 mV s^{-1} and a step size of 2 mV.

Fig. S 3. LSV curves of LPAac and Lac.

The acid retention test (DOI:10.1016/J.POLYMER.2015.03.040) involved suspending the PA-doped membranes above boiling water for 5 hours. The weight of the membranes was recorded every hour to measure the amount of acid leached. The time dependence of weight loss ratio of acid in different membranes is shown in Fig.S10.

Fig. S 10. Time dependence of weight loss ratio of acid in PBI, one side laser treated PBI and two side laser treated PBI.

REVIEWER COMMENTS

Reviewer #1 (Remarks to the Author):

I have no more comments.

Reviewer #2 (Remarks to the Author):

The authors have made improvements to the manuscript in this version; however, the following concerns still need to be addressed:

1. Based on the statements in the abstract and conclusion, it seems that the key finding of this work is the optimization of the three-phase interface through laser treatment. However, the issue of leaching has not been mentioned. Anyway, the unique contribution of this work remains unclear and should be clearly articulated and emphasized in the abstract.
2. The electrochemical performance of the MEA based on PBI after PA 'doping' without laser processing should be provided for comparison. This is crucial to assess the impact of laser processing.
3. EDX and Raman analysis alone cannot confirm the formation of graphene oxide; they only indicate the presence of oxygen-containing carbon. Direct evidence, such as TEM observations of layered graphene oxide, is needed to substantiate this claim.
4. At a minimum, a long-term stability test of the optimal MEA based on LPAC should be conducted for at least 100 hours to determine whether the performance can be further improved or stabilized.
5. Is PA leaching a major factor affecting MEA performance? While two-sided laser treatment appears more stable than one-sided treatment as shown in Fig. S10, the MEA performance of LPAC is superior to that of LPAac in Fig. 2. This discrepancy should be clarified.
6. The test duration shown in Fig. S10 is insufficient. I recommend extending the test to at least 12 hours.

Reviewer #3 (Remarks to the Author):

The authors addressed my comments. I would recommend the acceptance of this manuscript.

We would first like to thank the reviewer and editors for their further input, we greatly appreciate their additional feedback. We have revised the manuscript in line with the comments provided. Our responses to the questions raised can be found below.

Additions to the manuscript have been highlighted (see highlighted MS).

The authors have made improvements to the manuscript in this version; however, the following concerns still need to be addressed:

1. Based on the statements in the abstract and conclusion, it seems that the key finding of this work is the optimization of the three-phase interface through laser treatment. However, the issue of leaching has not been mentioned. Anyway, the unique contribution of this work remains unclear and should be clearly articulated and emphasized in the abstract.

We have now rewritten the abstract to better reflect the key focus of the manuscript:

High-temperature proton exchange membrane fuel cells (HT-PEMFCs) offer solutions to challenges intrinsic to low-temperature PEMFCs, such as complex water management, fuel inflexibility, and thermal integration. However, they are hindered by phosphoric acid (PA) leaching and catalyst migration, which destabilize the critical three-phase interface within the membrane electrode assembly (MEA). This study presents an innovative approach to enhance HT-PEMFC performance through membrane modification using picosecond laser scribing, which optimises the three-phase interface by forming a graphene-like structure that mitigates PA leaching. Our results demonstrate that laser-induced modification of PA-doped membranes, particularly on the cathode side, significantly enhances the performance and durability of HT-PEMFCs, achieving a peak power density of 817.2 mW cm^{-2} after accelerated stress testing, representing a notable 58.2% increase compared to untreated membranes. Furthermore, a comprehensive three-dimensional multi-physics model, based on X-ray micro-computed tomography data, was employed to visualise and quantify the impact of this laser treatment on the dynamic electrochemical processes within the MEA. Hence, this work provides both a scalable methodology to stabilise an important future membrane technology, and a clear mechanistic understanding of how this targeted laser modification acts to optimise the three-phase interface of HT-PEMFCs, which can have impact across a wide array of applications.

2. The electrochemical performance of the MEA based on PBI after PA 'doping' without laser processing should be provided for comparison. This is crucial to assess the impact of laser processing.

We apologise to the reviewer for the confusion caused. In high-temperature proton exchange membrane fuel cells, proton conduction relies on the Grotthuss

mechanism, which requires a hydrogen bond network established between PBI and PA. Dry PBI alone cannot facilitate proton conduction, as illustrated in the schematic diagram below.

Therefore, PA doping of the membrane is essential in HT-PEMFCs. Hence, all membranes discussed in the manuscript are PA-doped, with the distinction between laser-treated membranes lying in whether the laser treatment was applied before or after PA doping. We had previously used “PBI” alone to represent PA-doped PBI, but we have now revised the legend in Figure 2, changing “PBI” to “PBI/PA”.

Fig. 1. Electrochemical characterization of MEAs based on different membranes 160 °C, anode: H₂ ($\lambda=1.2$), cathode: O₂ ($\lambda=2.0$). **a** AST (repeating chronopotentiometry between 0.6 A cm⁻² for 4 min and 1 A cm⁻² for 16 min, running OCV for 10 min every 6 hr), **b** polarization curves before AST, **c** polarization curves after AST, **d** EIS curves before AST (Nyquist plots, DC: 0.5 A cm⁻², frequency:10 kHz-0.1 Hz), **e** EIS curves after AST (Nyquist plots, DC: 0.5 A cm⁻², frequency:10 kHz-0.1 Hz)

3. EDX and Raman analysis alone cannot confirm the formation of graphene oxide; they only indicate the presence of oxygen-containing carbon. Direct evidence, such as TEM observations of layered graphene oxide, is needed to substantiate this claim.

Due to the fact that this work involves the modification of only a thin surface layer on the membrane, generating a thin film of graphene oxide (GO), it is challenging to separate the GO formed on the surface for TEM analysis. However, we acknowledge that, as the reviewer mentioned, the D and G peak positions of GO and graphite oxide in Raman spectroscopy are nearly identical, and although their characteristics differ slightly, Raman spectroscopy under the conditions present in PBI may not be sufficient to distinctly identify GO alone. We have hence supplemented the study with X-ray diffraction (XRD) analysis. Grazing-incidence X-ray diffraction (GIXRD) data were collected by Malvern Panalytical Empyrean X-ray diffractometer. The radiation source was Cu K α ($\lambda = 1.5406\text{\AA}$) operated at 40kV, 40mA with an incidence angle $\omega = 1^\circ$. All patterns were obtained by using a step scan method (0.05° per step for 1s), in a 2θ range from 5° to 60° .

(XRD is a powerful technique used to characterize and differentiate graphite, graphene oxide (GO), and graphene based on their unique structural features:

1. **Graphite:** Graphite typically shows a strong and sharp peak around $2\theta \approx 26.5^\circ$, corresponding to the (002) plane, which is indicative of its highly ordered, layered structure. This peak is a characteristic of the well-stacked graphene layers in graphite.
2. **Graphene Oxide (GO):** GO exhibits a distinct peak at around $2\theta \approx 10\text{--}12^\circ$. This shift to a lower angle compared to graphite is due to the increased interlayer spacing caused by the presence of oxygen-containing functional groups such as hydroxyl, epoxide, and carboxyl groups. The increased spacing disrupts the regular stacking order seen in graphite.
3. **Graphene:** Graphene, especially when few-layered or single-layered, often exhibits a broad and weak peak around $2\theta \approx 24\text{--}26^\circ$. This peak is typically much less intense compared to graphite due to the reduced number of layers and less ordering. In many cases, XRD peaks for single or few-layer graphene might be weak or even absent due to the limited periodicity in the vertical direction (c-axis).

The differences between various graphite and graphene-based materials can be visually demonstrated in the figure above (DOI: [10.13005/ojc/340120](https://doi.org/10.13005/ojc/340120)). These distinct XRD patterns allow researchers to differentiate between graphite, GO, and graphene by examining peak positions, intensities, and the degree of ordering or disorder within the materials.

As shown in Figure 1e (ii) of the manuscript, compared to PBI, the laser-induced PBI displays a characteristic peak of GO at $2\theta = 11.68^\circ$, providing additional evidence of the formation and presence of GO. We believe that this result, combined with the aforementioned Raman analyses, substantiates the successful generation of GO on the membrane surface.

Fig. 1. Laser-induced membrane. e, (i) Raman spectra of PBI membrane and Laser-induced PBI membrane, (ii) XRD spectra of PBI membrane and Laser-induced PBI membrane.

4. At a minimum, a long-term stability test of the optimal MEA based on LPAc should be conducted for at least 100 hours to determine whether the performance can be further improved or stabilized.

As requested we have included AST analysis of the LPAc ranging from 37 hours to 111 hours. This analysis indicates that LPAc exhibits highly stable performance in the later stages of AST.

Fig. S 5. Extended AST plots of LPAc, (repeating chronopotentiometry between 0.6 A cm^{-2} for 4 min and 1 A cm^{-2} for 16 min, running OCV for 10 min every 6 hr)

5. Is PA leaching a major factor affecting MEA performance? While two-sided laser treatment appears more stable than one-sided treatment as shown in Fig. S10, the MEA performance of LPAc is superior to that of LPAac in Fig. 2. This discrepancy should be clarified.

Fig. S10 (now updated to Fig. S11) illustrates the impact of laser induction on the rate of PA leaching. However, we demonstrate that the influence of PA leaching on HT-PEMFC performance is complex and multifaceted, as evidenced by a combination of electrochemical tests, CT scanning, and multiphase multi-physics simulations.

In HT-PEMFCs, a moderate amount of PA leaching into the catalyst layer is necessary to facilitate proton conduction within the CL. However, excessive PA leaching can result in the complete coverage of the catalyst layer, impairing

performance. Single-sided and double-sided laser treatments induce distinct changes in the membrane morphology, catalyst structure, and mechanical properties, leading to variations in PA leaching behaviour. Thus, the impact of PA leaching on HT-PEMFCs is not solely determined by the amount or rate of PA leaching but also by the resulting structural and mechanical changes in the membrane and catalyst layer.

For instance, LPAac exhibits a smooth membrane and pristine MEA electrode structure, which might initially appear ideal. However, this resembles the state of a non-activated MEA. HT-PEMFCs have unique characteristics: the catalyst layers do not use ionomers but PTFE as a binder, and effective proton conduction requires some level of PA infiltration. The high oxygen and hydrogen flux observed in LPAac suggests that the CL contains minimal PA, limiting the propagation of reactions beyond the catalyst surface. This explains why, despite efficient material transport, the charge and mass transfer resistance ($R_{\Delta(c+m)}$) in LPAac remains high. Additionally, laser treatment slightly reduces proton conductivity, with a more pronounced effect when both sides are treated compared to single-sided treatment.

This study underscores that the superior performance of HT-PEMFCs does not arise from an ostensibly “perfect” three-dimensional structure or the optimization of a single phase. Instead, optimal performance is achieved through the complex interplay and synergy of multiple phases and physical fields, highlighting the advantages of seemingly "imperfect" structures.

6. The test duration shown in Fig. S10 is insufficient. I recommend extending the test to at least 12 hours.

The test shown in Fig. S10 (now updated to S11) was extended to 12 hours, as depicted below.

Fig. S 11. Time dependence of weight loss ratio of acid in PBI, one side laser treated PBI and two side laser treated PBI.

REVIEWERS' COMMENTS

Reviewer #2 (Remarks to the Author):

The authors have made proper revision in this version. This manuscript could be accepted as it is.